# CubeDiff: Repurposing Diffusion-Based Image Models for Panorama Generation

**Nikolai Kalischek**
ETH Zürich, Google

**Michael Oechsle**
Google

**Fabian Manhardt**
Google

**Philipp Henzler**
Google

**Konrad Schindler**
ETH Zürich

**Federico Tombari**
Google

## Abstract

We introduce a novel method for generating 360° panoramas from text prompts or images. Our approach leverages recent advances in 3D generation by employing multi-view diffusion models to jointly synthesize the six faces of a cubemap. Unlike previous methods that rely on processing equirectangular projections or autoregressive generation, our method treats each face as a standard perspective image, simplifying the generation process and enabling the use of existing multi-view diffusion models. We demonstrate that these models can be adapted to produce high-quality cubemaps without requiring correspondence-aware attention layers. Our model allows for fine-grained text control, generates high resolution panorama images and generalizes well beyond its training set, whilst achieving state-of-the-art results, both qualitatively and quantitatively. Project page: https://cubediff.github.io/

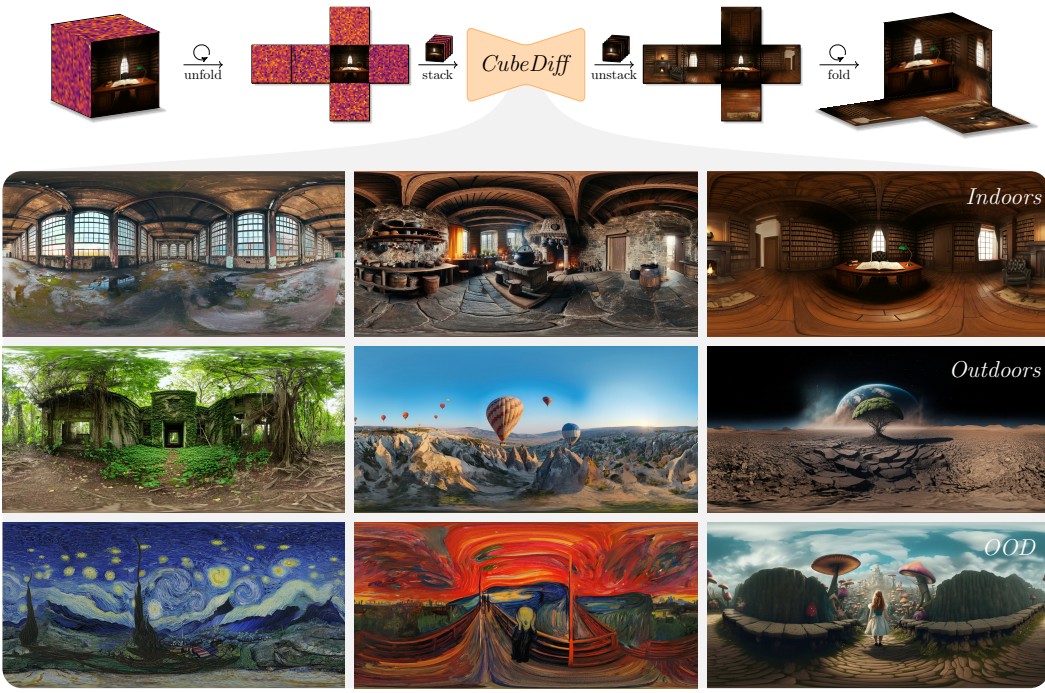

Figure 1: *CubeDiff* leverages cubmaps to represent 360° panoramas and denoises all faces together in a single pass. In contrast to other works, *Cubediff* does not need to consider distortions, since it operatkes on common 90° FOV perspective images, maing it possible to directly utilize the internet-scale image prior of the underlying diffusion model.

# 1 INTRODUCTION

Recent advances in diffusion-based generative models have seen tremendous progress over the last two years, enabling a wide range of applications from artistic expression and product design to personalized content creation. Beyond generating realistic and diverse images based on text-to-image models (Rombach et al., 2022; Saharia et al., 2022), these models are now capable of more complex tasks such as 3D asset creation (e.g., (Kalischek et al., 2022; Wang et al., 2024; Mohammad Khalid et al., 2022; Poole et al., 2022)), estimating scene properties such as depth or semantics (Ke et al., 2024; Baranchuk et al., 2021), illumination changes (Jin et al., 2024; Zhao et al., 2024; Zeng et al., 2024), and generation of multi-view consistent images (Gao et al., 2024b; Tang et al., 2023).

The latter is particularly interesting in virtual reality, gaming and entertainment, where 3D consistency is crucial for fully immersive experiences and thus user satisfaction. However, synthesizing high-quality, visually coherent panoramas presents unique challenges. First, capturing sufficient panoramic data is tedious and costly, as specialized cameras and/or additional processing are needed to remove stitching artifacts. Consequently, models must be trained in a low-data regime making them prone to overfitting, this limiting their generalization capabilities. Exemplary, a lot of models are restricted to indoor environments only (Wu et al., 2023; Song et al., 2023). Second, panoramas must fulfill additional constraints compared to perspective images. Most notably, the image borders must align to allow a seamless wrap-around. But there are also more intricate, semantic constraints, e.g., the viewing frustum must cover the entire scene. Hence, when generating a panorama of a bedroom, it must contain *exactly one* bed, *at least one* door, etc. On the other hand, outdoor panoramas should maintain realistic spatial relationships between elements.

To satisfy those requirements, prior work had to introduce complex additional model components (Gao et al., 2024a; Tang et al., 2023; Yang et al., 2024), or employ dedicated mechanisms such as autoregressive outpainting from a perspective view (causing artifacts like content drift and the Janus effect (Wang et al., 2023), and circular padding to enforce consistent wrap-around (Feng et al., 2023; Wu et al., 2023).

We introduce a simple yet highly effective solution: we generate panoramas using a fine-tuned multi-view diffusion model, following recent line of work (Gao et al., 2024b; Tang et al., 2023; Zhang et al., 2023b). This approach leverages the inherent properties of cubemaps, where a 360° × 180° panorama is represented by six perspective images on the faces of a cube. This allows us to fully recycle a pretrained text-to-image model, enabling generalization far beyond the limited training data. Contrary to existing methods, the architectural modifications we require to ensure consistency between cube faces are minimal: all attention layers are inflated by one additional dimension to enable crosstalk between the six faces. This simple modification, combined with fine-tuning on panorama data, achieves state-of-the-art results with significant visual and semantic coherence. Additionally, the model allows for fine-grained text control by training with face-specific image-text pairs, easily generated by prompting an LLM to produce per face text descriptions.

Our key insight is that existing, generative image models can be easily extended to generate high-resolution panoramas, by performing diffusion in cubemap space and adding attention mechanisms to other faces within the cubemap, see Figure 1. The resulting model

- enables consistent image generation across all cubemap faces and preserves the internet-scale image prior of the underlying diffusion model to generalize beyond the training panoramas;
- delivers state-of-the-art results on panorama generation, both qualitatively and quantitatively, and outperforms previous methods in terms of visual fidelity and coherence;
- enables efficient high-resolution synthesis, benefiting from current and future advances in off-the-shelf image diffusion models;
- allows for novel fine-grained text control, enabling users to guide the generation with detailed textual descriptions.

# 2 RELATED WORK

Similar to 3D generative modelling, training data for panorama generation is scarce and much effort has been spent on how to repurpose standard perspective image priors for panoramas. The preva-

lent approach has been to autoregressively outpaint panoramas, more recently multi-view diffusion models have attracted interest. We now discuss relevant works and differences to our approach.

## 2.1 PANORAMA GENERATION.

Most panorama generators operate in equirectangular projection, thus having to deal with it severe nonlinear distortions (especially near the poles). Previous methods either autoregressively outpaint the panorama (Gao et al., 2024a; Lu et al., 2024; Wang et al., 2023) or generate the entire equirectangular image in one shot (Feng et al., 2023; Wu et al., 2023). They are commonly conditioned on either a single narrow field-of-view image (Akimoto et al., 2022) or solely on a text prompt (Chen et al., 2022). The state of the art are diffusion models, which have gradually replaced adversarial approaches (Akimoto et al., 2022; Somanath & Kurz, 2021). Feng et al. (2023) fine-tune a latent diffusion model on a panorama dataset and apply a circular blending strategy in the denoising and decoding stages to enforce consistent wrap-around. Similarly, Wu et al. (2023) stitches the right part of the image to the left part in latent space in each denoising step. Such blending improves the results, but encumbers the inference step. In our method it is not required. Lu et al. (2024) propose to autoregressively outpaint a panorama with a complex architecture of submodules for panorama-aware visual guidance, NFoV guidance and panorama-aware geometric guidance. In Wang et al. (2023), the authors extend the outpainting task to ingest multiple NFoV images of the same scene. A two-stage network predicts their relative rotations, then a diffusion model with ControlNet (Zhang et al., 2023a) outpaints the panorama based on the projected inputs. Recently (Voynov et al., 2023) introduce a diffusion model with control over the rendering geometry, including panoramic outputs. Gao et al. (2024a) additionally incorporate a state space model to aggregate global information into cross-attention layers of the diffusion model, building up the panorama by inpainting empty regions. The present work demonstrates that, with the right representation, high-quality panoramas can be obtained without inflating the complexity and brittleness of the architecture. Related to panorama generation is the more modest strategy to alter existing panoramas by injecting a user-defined style, in either equirectangular or cubemap projection (Yang et al., 2024; Song et al., 2023).

## 2.2 MULTI-VIEW DIFFUSION

Multi-view diffusion models offer a compelling alternative to equirectangular or autoregressive panorama generators. Zhang et al. (2023b) introduces a compositional diffusion scheme that enables the generation of large-scale content, leveraging models trained on smaller constituent parts. That work is based on factor graphs, and demonstrates how the cubemap can be turned into a factor graph in order to train a diffusion model conditioned on segmentation maps. The work most closely related to ours is Tang et al. (2023). It aims to generate cylindrical panoramas (i.e., 360°horizontal field of view, but restricted vertical view angle). The authors propose a sophisticated correspondence-aware cross-attention between local neighborhoods of eight perspective feature maps spaced at 45°angles. Recently, Gao et al. (2024b) and Shi et al. (2023) discovered that expanded attention layers that connect not only features within an image but also across multiple images, are beneficial when handling multiple object-centric views. Our approach turns this setup inside-out and extends a pretrained text-to-image (T2I) model in a similar manner for panorama generation. We instead do not require camera pose or 3D information, due to the fixed viewing geometry of the cubemap.

## 3 PANORAMA REPRESENTATIONS

Panoramic images aim to capture a complete $360° \times 180°$ view of a scene from a fixed view point. There exist several different panorama representations in literature, each with its own advantages and drawbacks. This section briefly discusses the most prominent ones.

**Spherical projection** maps a 360° view onto a sphere, preserving the geometric relationships between points in the scene. Points are generally defined using longitude and latitude. While conceptually intuitive, directly utilizing a spherical representation for image processing is challenging due to difficulties in representing a sphere on a flat image plane, which often leads to distortions and non-uniform sampling densities in practical implementations.

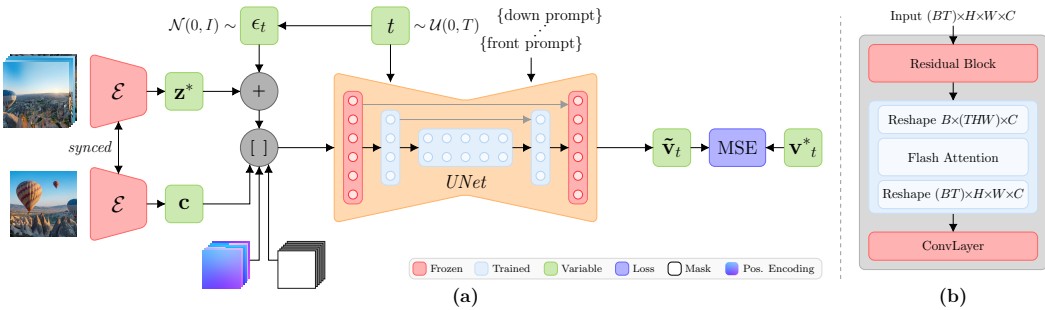

Figure 2: **An overview of our training pipeline and panorama model. (a)** We project all training panoramas onto a cubmap and feed the faces to our frozen VAE encoder with synchronized Group-Norm to obtain the respective latents and enrich them with panorama-specific positional encodings for explicit spatial awareness. **(b)** We only train the inflated attention layers to be cross-frame aware.

**Equirectangular projection** projects the spherical panorama onto a 2D rectangle. To this end, latitude and longitude coordinates on the sphere are mapped to vertical and horizontal coordinates on a rectangle. While widely used due to its simplicity, equirectangular projection suffers from significant distortions, especially near the poles where horizontal stretching becomes extreme. This distortion affects both visual quality and the performance of algorithms processing equirectangular panoramas, as most existing T2I models process images with NFoV images.

**Cubemaps** offer an alternative representation where a 360° view is projected onto the six faces of a cube. Each face captures a 90° field of view, resulting in six perspective images that can be seamlessly stitched together. This representation avoids the polar distortions inherent to equirectangular projections, providing more uniform sampling, making it highly applicable to existing diffusion models trained on vast amount of perspective images. However, note that cubemaps introduce discontinuities at the edges of the cube faces, which needs to be handled carefully.

## 4 METHOD

We introduce *CubeDiff*, a novel approach for generating high-quality, consistent panoramas using the cubemap representation. *CubeDiff* generates the six perspective views of a cubemap in parallel and context-aware manner, exploiting the strengths of pretrained T2I diffusion models. Below, we delve into the architectural choices that enable *CubeDiff* to achieve high-quality and consistent panoramas, while retaining strong generalization capabilities inherited from the pretrained model. Similar to Gao et al. (2024b), *CubeDiff* comprises a variational autoencoder (VAE) and a latent diffusion model (LDM), mirroring the structure of conventional T2I diffusion models. However, we carefully adapt each component for effective multi-view panorama generation.

### 4.1 MODEL ARCHITECTURE

The latents produced by the VAE are used to fine-tune a pretrained LDM operating on a 128x128x8 latent space, initialized with weights from a model trained on a large-scale image dataset. The pretrained LDM consists of an architecture similar to Stable Diffusion (Rombach et al., 2022), which is build with multiple convolutional, self-attention, and cross-attention layers. To enable cross-view awareness and maintain global consistency, we inflate all existing 2D attention layers, *i.e.* both self-attention and cross-attention for text conditioning. These layers, adapted from (Shi et al., 2023), extend the attention mechanism across all six cube faces, allowing the model to learn relationships and dependencies between different viewpoints. Inflating of layers can be easily conducted by extending the token sequence length from $b \times (hw) \times l$ to $b \times (thw) \times l$, *e.g.* for self-attention, where $b$ is the batch size, $hw$ the flattened spatial size and $t = 6$ the cube length. While this is different to more sophisticated attention layers (Tang et al., 2023; Huang et al., 2024), it in turn enables us to retain the original pretrained attention weights, which reduces the risk of overfitting and thus greatly improves overall performance.

The LDM receives two conditioning signals. We incorporate text embeddings, either one common prompt or one prompt for each face, and a single conditional view of the scene (w.l.o.g. we assume the front face of the cube). During training, we concatenate the VAE latents of the conditioning views to the noisy latents of the target views, providing the LDM with complete context information. Furthermore, we incorporate a binary mask channel into the latent representations. This mask distinguishes between conditioning views (provided as clean latents) and target views (subjected to noise injection during training). We show an overview of our model architecture in Figure 2.

## 4.2 SYNCHRONIZED GROUPNORM

Our VAE architecture incorporates synchronized group normalization, a crucial element for achieving consistent color tones across the generated panorama. Since our VAE processes the six faces of a cubemap as a batch of six individual images, standard group normalization can lead to subtle color inconsistencies among different views (*c.f.* fig. 6a). This occurs as feature statistics are computed and normalized independently for each image in the batch. Without synchronization, encoding and decoding a panorama results in noticeable shifts, particularly evident in the equirectangular projection. Synchronized group normalization addresses this issue by jointly normalizing feature activations across both spatial and inter-view dimensions. Consequently, synchronized group normalization contributes significantly to the generation of visually harmonious and coherent panoramas. Similar effects have been observed in (He et al., 2023). We further discuss this in Section 5.6 and compare synchronized and unsynchronized results in Figure 6a.

## 4.3 POSITIONAL ENCODING

To provide the LDM with explicit spatial awareness within the cubemap, we augment the latent representations with positional encodings derived from the 3D geometry of the cube. For each point on a cube face, we compute its corresponding UV coordinates on the unit cube, defined by:

$$u = \arctan 2(x, z) \quad , \quad v = \arctan 2(y, \sqrt{x^2 + z^2}), \tag{1}$$

where $(x, y, z)$ are the 3D coordinates of the point on the cube face, projected onto the unit cube. These UV coordinates are then normalized to $[0, 1]$ and concatenated as two additional channels to the (noisy) latents. This positional encoding scheme provides the model with information about the spatial location of each latent patch within its respective cube face, facilitating the generation of panoramas with consistent geometry and object relationships across views.

## 4.4 OVERLAPPING PREDICTIONS

To further enhance the geometric and color consistency across cube faces, we introduce overlapping predictions during both training and generation. Instead of generating each face with a 90° field of view (FoV), we enlarge the FoV by 2.5° on each side, resulting in an effective FoV of 95° per face. This means each generated face includes a small overlap with its neighboring ones. This overlapping generation strategy serves two purposes. During training, it encourages the model to learn consistent representations across adjacent faces, as the overlapping regions provide additional context and constraints. During panorama assembly, we discard these overlapping regions and only retain the central 90° portion of each generated face. This strategy effectively avoids the need for explicit blending operations at the cube face boundaries, which can sometimes introduce subtle artifacts. The overlaps can be seen at the boundaries of the cubemaps in Figure 3 (*e.g.*, the duplicated fireplace in the right and back views) and in the appendix.

## 4.5 CLASSIFIER-FREE GUIDANCE

We employ classifier-free guidance (CFG) (Ho & Salimans, 2022) on both the text and image conditions during training. Thereby, we randomly drop either the text prompt, the conditional image, or both. When the text prompt is dropped, it is replaced with null tokens in the cross-attention layers; when the conditional image is dropped, its corresponding tokens in the self-attention layers are masked out by setting them to negative infinity, effectively zeroing out their attention weights. This training procedure enables diverse panorama generation scenarios during inference. Users can provide both text and image conditions for maximum control and fidelity or drop both or either condition to explore unconditional generation modes.

## 5 EXPERIMENTS

This section details our experimental setup, followed by quantitative and qualitative evaluations. We compare the performance of *CubeDiff* against the state-of-the-art and ablate our design choices.

### 5.1 EVALUATION PROTOCOL

**Training and inference setup**    We finetune our model using Adam (Kingma & Ba, 2014) and train for 30,000 iterations with batch size 64. The learning rate is ramped up to $8 \times 10^{-5}$ in the first 10,000 steps. During training, we employ classifier-free guidance, dropping conditional signals 10% of the time. We find it important to not only drop the text condition in the cross-attention layers but to also zero out the input condition in the self-attention layers. The diffusion model is finetuned using v-prediction (Salimans & Ho, 2022). We employ DDIM sampling (Song et al., 2020) with 50 steps during inference.

**Datasets**    *Training.*   We train on a mixture of indoor and outdoor environments by combining multiple publicly available sources, including Polyhaven (polyhaven.com, accessed 09/2024), Humus (Persson, accessed 09/2024), Structured3D (Zheng et al., 2020) and Pano360 Kocabas et al. (2021), giving in total around 48000 panoramas for training. While Humus provides an explicit cubemap representations, all other datasets come with equirectangular panoramas. We thus first generate cubemaps from these panoramas using standard perspective projection, ensuring consistent overlap between adjacent faces. To further enable text-guided panorama generation, we infer textual descriptions for each panorama in the datasets using the publicly available Gemini model (Gemini Team Google, 2023). We explore two captioning strategies: (1) generating a single caption for the entire panorama by providing Gemini with all six cube faces as input and (2) generating individual captions for each face independently, enabling fine-grained text control.

*Testing.*   We evaluate our method on the common Laval Indoor (Gardner et al., 2017) and Sun360 (Xiao et al., 2018) datasets. Laval Indoor consists of over 2100 high quality panorama captures of various indoor environments, Sun360 encompasses around 1000 panoramas including both – indoor and outdoor scenes. Note that we use those datasets only for evaluation, while Diffusion360 also uses Sun360 for training and OmniDreamer even leverages both datasets to train their models. Nonetheless, we decided to use these datasets for the sake of fairness and due to the lack of any proper overlapping test datasets.

**Metrics**    We use various metrics and modalities for evaluation – including perceptual metrics, text alignment, and a user study.

*Perceptual Metrics.* We use the very common Fréchet Inception Distance (FID) (Heusel et al., 2017) metric to measure the similarity between the distribution of real and generated images in a feature space derived from a pretrained Inception network. Lower FID scores indicate greater similarity and, thus, higher image realism; We additionally report the CLIP-FID (Kynkäänniemi et al., 2022) metric, replacing the Inception network with CLIP (Radford et al., 2021) to leverage its semantic understanding capabilities through a joint image-text embedding space. This metric captures thus both – visual fidelity and text-image alignment; Finally, we employ the kernel inception distance (KID)(Bińkowski et al., 2018). Similar to FID, KID uses features from a pre-trained network, however, it quantifies the difference between real and generated data distributions using the maximum mean discrepancy rather than the Fréchet distance.

*Text Alignment.* To measure text alignment we refer to the common CLIP score (Hessel et al., 2021) . The CLIP score computes the cosine similarity within the shared text-image embedding to measure the agreement between generated panoramas and their corresponding text prompts. Hence, a higher CLIP score indicates stronger semantic agreement between image and text.

**Competitors**    We compare *CubeDiff* to various state-of-the-art panorama generation methods. As for plain text to panorama generation, we employ Text2Light (Chen et al., 2022) and PanFusion (Zhang et al., 2024) to serve as our main competitors. For single image conditioning, we respectively use OmniDreamer (Lu et al., 2024) and PanoDiffusion Wu et al. (2023) as representatives for autoregressive and direct panorama generation based approaches. Finally, we compare against

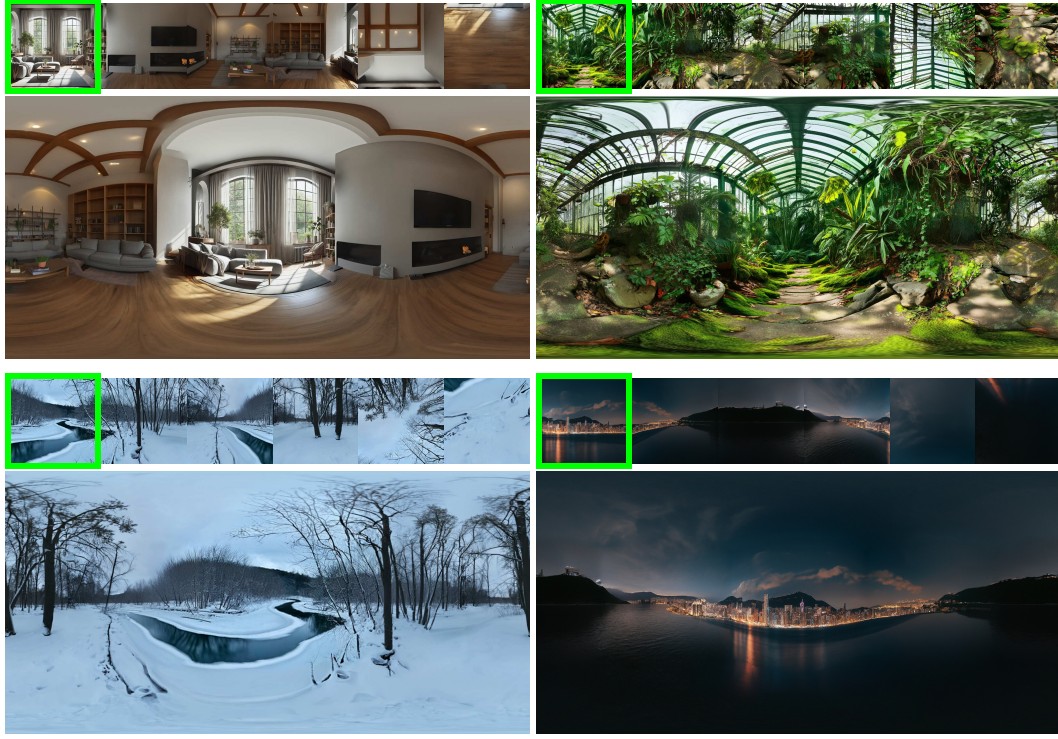

Figure 3: **Cubemaps and panoramas generated by *CubeDiff* with image and text condition.** We depict a diverse set of generated panoramas including indoor, outdoors, bright and dark scenes. In all settings, *CubeDiff* produces high quality and realistic panoramas that align with the input image.

Diffusion360 (Feng et al., 2023) and MVDiffusion (Tang et al., 2023) for text and image condition-ing based methods. Note that while Diffusion360 directly outputs panorama images, MVDiffusion instead employs multi-view diffusion models with a custom cross attention mechanism. Overall, the choice of baselines represents a variety of different generation techniques, covering various different tasks. Please note that none of the existing methods besides MVDiffusion offers the possibility to condition specific parts of the panorama on individual text prompts.

## 5.2 QUALITATIVE EVALUATION

In this section, we provide a qualitative evaluation of our method. We first present several conditional image generations of our method, before comparing *CubeDiff* against the state-of-the-art.

**Conditional image generation.** In Figure 3, we show generated panoramas given text-image pairs as condition. We considered input conditions that cover a broad range of scenes, such as outdoor and indoor scenes, bright and dark settings as well as texture rich and uniformly colored areas. Note that we do not show the text conditions due to limited space, however, we provide them in the appendix. We see that our approach yields high quality results under these diverse input settings. We especially emphasize the level of detail and geometric consistency beyond the input image.

**Qualitative comparison.** For visual comparison against the state-of-the-art, we show generated panoramas and their respective perspective projections in Figure 4. To this end, we sample random image and text pairs from the LAVAL Indoor dataset. We further group the methods according to their input modalities. Compared to the text-only approach Text2Light, our method is able to produce much more complex panoramas with better details and visual appeal. As for image-only approaches, we see that *CubeDiff* is capable of producing the most realistic panoramas. In particu-lar, while OmniDreamer suffers from blurry regions, PanoDiffusion is not able to properly transfer

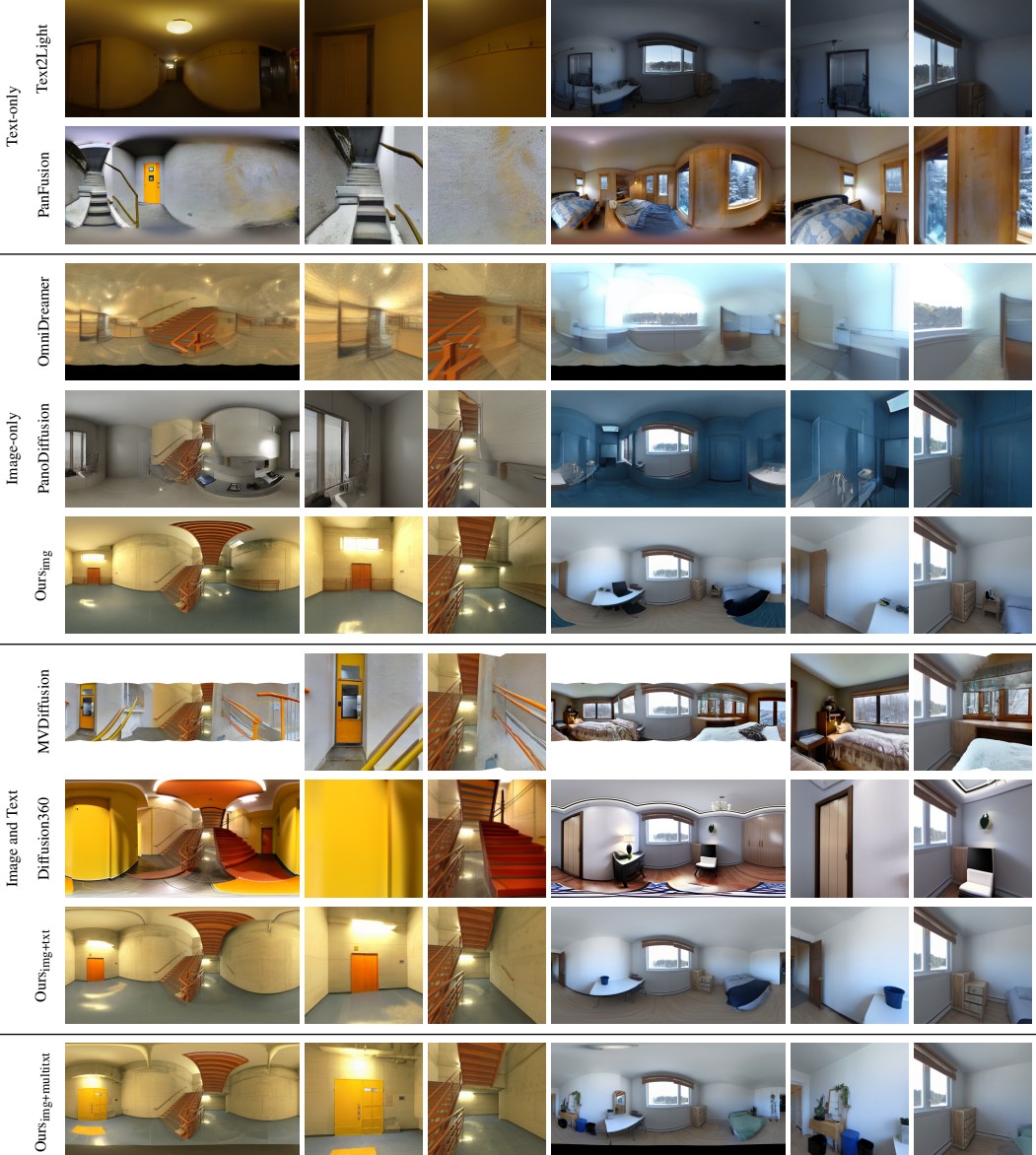

Figure 4: **Qualitative comparison between *CubeDiff* and baselines on the LAVAL Indoor Dataset.** Besides Text2Light, all panoramas are generated using the center face as input condition and additional text prompts if applicable. For each sample we show the panorama image as well as two projected images. Please zoom in to compare the different methods.

the input image appearance across the whole panorama. Finally, also for text and image conditioning our method again produces the best results, especially in terms of geometry. For example, while MVDiffusion is indeed capable of generating high quality images, the method sometimes produces inaccurate geometries as, for example, some walls and hand rails exhibit bending artifacts after perspective projection. Similarly, Diffusion360 occasionally suffers from implausible indoor layouts. To summarize, despite of using different input modalities, *CubeDiff* always generates high quality panoramas, surpassing all other state-of-the-art works in terms of visual appeal and geometric consistency.

| | LAVAL Indoor | | | | | SUN360 | | | | |
|---|---|---|---|---|---|---|---|---|---|---|
| | FID ↓ | KID (×10²)↓ | Clip-FID ↓ | FAED ↓ | CS ↑ | FID↓ | KID (×10²)↓ | Clip-FID↓ | FAED ↓ | CS ↑ |
| Text2Light | 28.3 | 1.45 | 11.5 | 136.1 | 25.18 | 60.1 | 4.31 | 31.3 | 82.9 | 23.27 |
| PanFusion | 41.7 | 2.85 | 19.8 | 71.7 | 26.58 | 30.0 | 1.42 | 7.8 | 44.5 | 25.28 |
| OmniDreamer | 71.0 | 5.17 | 23.9 | 19.2 | - | 92.3 | 8.89 | 51.7 | 30.4 | - |
| PanoDiffusion | 58.6 | 4.08 | 26.6 | 106.8 | - | 52.9 | 3.51 | 28.9 | 98.0 | - |
| **Ours**$_{img}$ | 11.7 | 0.47 | 4.4 | 22.0 | - | 27.4 | 1.35 | 11.5 | 8.9 | - |
| Diffusion360 | 33.1 | 2.07 | 16.9 | 23.7 | 26.38 | 45.4 | 3.73 | 18.5 | 12.6 | 22.89 |
| **Ours**$_{img+txt}$ | **9.5** | **0.32** | **3.2** | **18.4** | 27.02 | 25.5 | **1.33** | 8.1 | 7.6 | 25.00 |
| MVDiffusion | 25.7 | 1.11 | 13.5 | - | 27.44 | 50.9 | 3.71 | 15.4 | 32.3 | 25.54 |
| **Ours**$_{img+multitxt}$ | 10.0 | 0.35 | 4.1 | 21.2 | **30.17** | **24.1** | **1.33** | **7.0** | **5.7** | **28.14** |

Table 1: **Quantitative Evaluation on the Laval Indoor and SUN360 dataset.** We provide a comparison to various competitors and different input modalities. The first block of rows are text-only methods, the second image-only, the third image and single text description and the last block are image and multi-caption methods.

## 5.3 QUANTITATIVE EVALUATION

In this section, we provide the results of our quantitative evaluation on the Laval Indoor and the SUN360 dataset. We evaluate all methods on perceptual quality and consistency.

In Table 1 we provide quantitative results for visual quality. Our method outperforms all competitors significantly, regardless of input modalities. For example, we can report a FID score of 9.47 on Laval Indoor, which is a 270% relative improvement compared to the second best performing method MVDIffusion, reporting a score of 25.7. Compared to works that use only image or text as input conditioning, the gap even widens with Text2Light and PanoDiffusion respectively reporting a FID of 28.3 and 58.6. This trend holds across all metrics. Interestingly, *CubeDiff* performs similarly across different input modalities, demonstrating its strong generalizibility.

However, the provided perceptual metrics can only evaluate the overall realism of the generated panoramas and are not capable of capturing consistency towards input. We next study the alignment to the input text prompt. To this end, we leverage the CLIP score to measure how well the generated panoramas align with the text input. As can be seen in the table our method surpasses the state-of-the-art again by a significant amount for all datasets and modalities, showing how precisely our model respects the textual input.

## 5.4 USER STUDY

We conducted a user study with a two-alternative forced choice (2AFC) survey to evaluate our panorama generation method. Each of the 28 participants was shown 30 pairs of generated panoramas alongside the original conditioning image and asked to select their preferred option based on quality, composition, style, and alignment with the condition image.

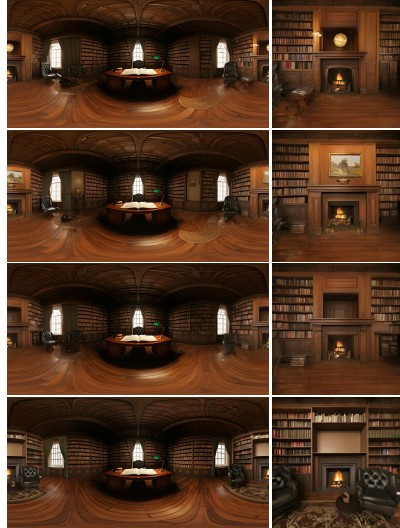

Figure 5: **Fine-grained Text Control.** We show an example for fine-grained-text control of the back face. Our model is able to change details following the provided prompt. First, we add a golden globe above the fireplace; second, we place a picture above the fireplace; third, we leave the space above empty; last, we instead add a bookshelf above it.

Our method outperformed competitors statistically ($p < 0.1$, binomial test). Specifically, 16.9%, 17.3%, and 19.5% of participants preferred our single-image, multi-image, and no-text variants, respectively. The no-text variant nearly matched the ground truth preference (19.9%), demonstrating our method's ability to generate realistic and accurate panoramas. In contrast, OmniDreamer, PanoDiffusion, MVDiffusion, and Diffusion360 had significantly lower preference rates of 1.7%, 5.3%, 7.0%, and 12.3%, respectively.

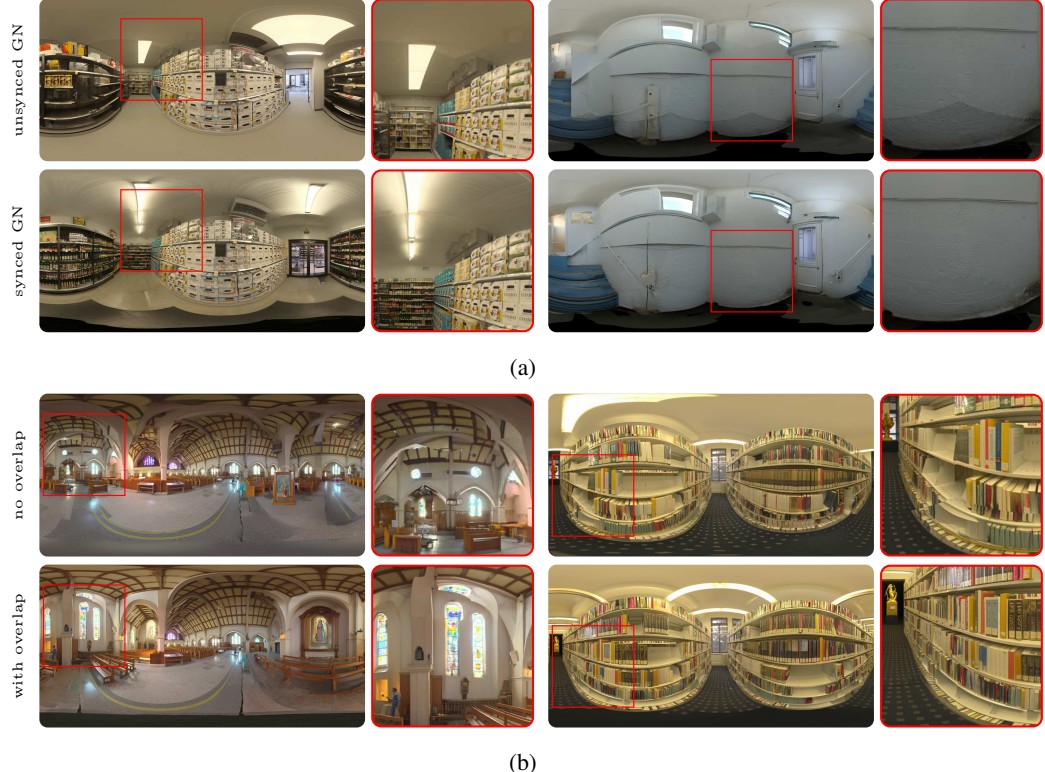

Figure 6: **Ablation on synchronized GN and overlap prediction.** **(a)** Top: Group normalization over the spatial dimension only. Bottom: Additional normalization over the frame dimension. **(b)** Top: Panoramas without overlapping cube faces. Bottom: Panoramas with our standard 2.5° overlap.

## 5.5 FINE-GRAINED TEXT CONTROL

Different to all competitors, our method enables complete fine-grained and per-face text control. For example in Figure 5, we show results for providing different text descriptions for the back face. We can always generate visually appealing results, regardless of the object we place above the fireplace.

## 5.6 ABLATIONS

**Synchronized Group Norm (GN)** Synchronized GN ensures consistency across cube faces by normalizing over both spatial and frame dimensions, as shown in Figure 6a. Without it, models often exhibit color inconsistencies and artifacts at cube face boundaries. While metrics like FAED may not capture these subtle issues, synchronized GN significantly improves visual quality.

**Overlapping Prediction** Overlapping predictions mitigate discontinuities at cube face boundaries by introducing small overlaps, as illustrated in Figure 6b. This ensures seamless transitions, with non-overlapping regions cropped for the final panorama. The approach leverages global context from full attention, eliminating visible seams without additional VAE finetuning.

## 6 CONCLUSION

This work introduces a novel approach to panorama generation leveraging pretrained text-to-image diffusion models applied to a cubemap representation. By enabling attention across the cube faces, our method achieves state-of-the-art results in terms of visual fidelity and coherence, while requiring minimal architectural changes. This approach not only inherits the strengths of existing diffusion models, including high-resolution synthesis and generalization capabilities, but also unlocks fine-grained text control over the generated panorama. This opens up exciting new possibilities for creative applications and paves the way for future research in controllable panorama generation.

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
