

Figure 1: **Panorama in a out of training-distribution setting.**

# A APPENDIX

## A.1 OUT-OF-DISTRIBUTION EXAMPLES

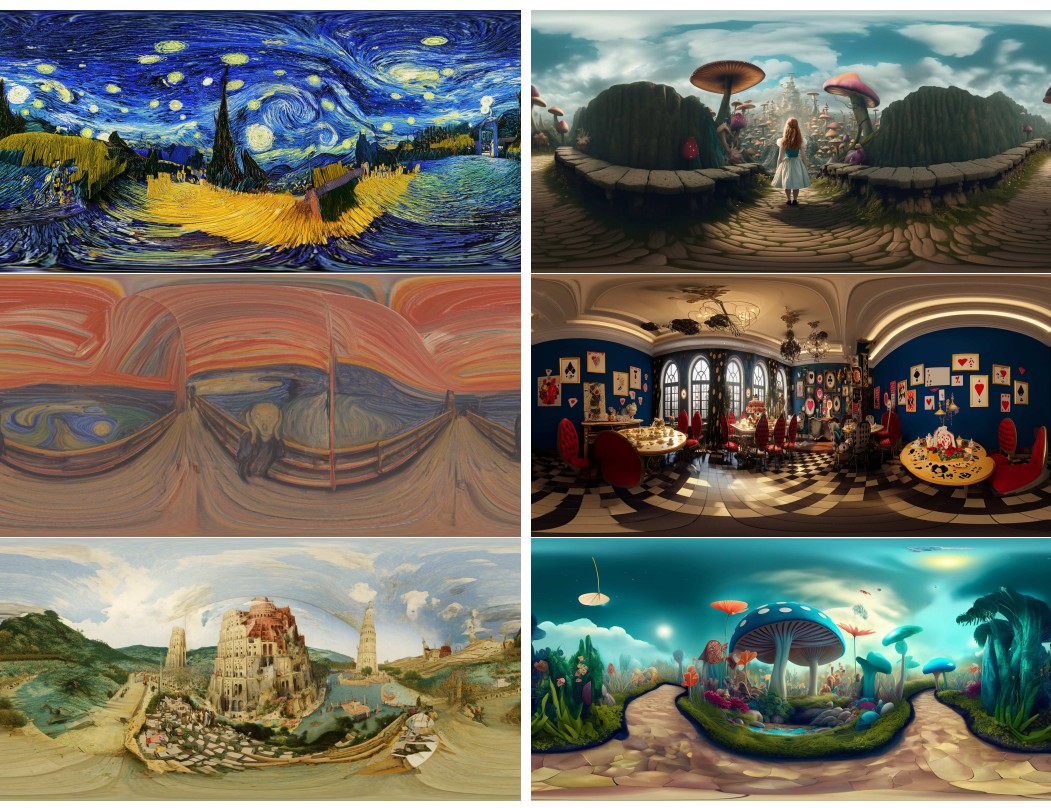

Figure 2: **Out-of-distribution examples generated by Cubediff.** On the right are artistic generations, on the left fantasy sceneries like Alice in Wonderland.

## A.2 Text prompt examples

The text prompts for the qualitative examples are the following:

**City skyline**

- A glittering cityscape at night, densely packed high-rise buildings illuminated against a dark sky. The buildings are various heights and designs, reflecting light onto the calm water in the foreground. A hill or mountain range forms a dark backdrop behind the city.

- Dark, silhouetted mountains stretching across the entire view, with the faintest glow of city lights on the horizon. The sky is a deep, nighttime blue.

- The side profile of a mountainous region, dark and slightly textured, extending from the foreground to the distant horizon. The city lights are visible in the distance to the right, providing a subtle contrast against the mountain's blackness.

- Similar to the right. A darker, less detailed section of the mountain range appears to the left.

- An expansive, dark night sky with subtle cloud formations. The upper portions of the city's tallest buildings are faintly visible as a horizontal line against the darkness.

- The dark, calm water of the bay reflecting the city lights, showing subtle ripples and disturbances. The reflection is most intense closest to the buildings and gradually fades into darkness.

**Green house**

- A dense, overgrown greenhouse teems with lush greenery, creating a vibrant, jungle-like scene. Large ferns, broad-leafed tropical plants, and hanging baskets overflowing with foliage dominate the view. Large rocks, blanketed in thick, *vibrant* green moss, suggest a path, nearly disappearing beneath the surrounding plants. The glass roof, tinged with green, casts a soft, diffused light.

- The right side of the greenhouse is a solid wall of plant life. Large, vibrant green leaves and hanging plants create a dense, tropical atmosphere. The rocks, almost entirely smothered in *bright* green moss and overflowing greenery, offer only the faintest hint of a path. Wild, untamed growth dominates the scene.

- The greenhouse extends into the distance, an endless expanse of green fading into shadow. The rocks, now completely obscured by plants and a thick carpet of *emerald* moss, suggest a path swallowed by the jungle. The sheer volume of greenery creates a sense of depth and wildness. The glass roof is barely visible.

- To the left, a tall palm tree rises amidst the dense foliage. Large ferns and other leafy plants create a solid wall of green around it. The rocks, barely discernible beneath the thick layer of *brilliant* green moss and creeping vines, continue on this side, obscuring the path almost entirely.

- Looking up, the glass panes of the greenhouse roof, tinted green with clinging vines and moss, are mostly obscured by the dense canopy. Hanging plants, heavy with ferns and leafy vines, cascade downwards, creating a lush, verdant ceiling. The metal framework is almost entirely hidden.

- Looking down, the large rocks are almost completely hidden beneath a thick carpet of *luminescent*, almost *glowing* green moss, fallen leaves, vines, and other plant debris. Only the edges of the rocks peek through the dense greenery, making the path nearly invisible. The texture of the moss appears incredibly soft and velvety, a seamless blanket of vibrant green.

**Living room**

- A cozy, modern living room with large windows allowing natural light to flood in. The room is furnished with a soft, gray sofa facing the windows and a wooden coffee table in

the center. A bookshelf filled with plants and books stands against the far wall. A fireplace in the corner crackles softly, casting a warm glow across the room.

- A side view of the living room from the right, showing the side of the gray sofa facing toward the large windows. The coffee table is positioned in front of the sofa. To the left, the wall-mounted television is visible above the fireplace, and the wooden floor stretches across the room. A small side table with a lamp sits next to the sofa.

- A view of the back wall of the room, where the bookshelf is the primary focus. The large windows let in a soft light, but the sofa and coffee table are out of sight from this angle. The fireplace is visible on the right side of the room, softly illuminating the space, and the television mounted above it is partially visible.

- A side view from the left side of the living room, showing the bookshelf along the far wall, and the curtains gently swaying in front of the large windows. The coffee table sits on a stylish rug, but the sofa itself is mostly out of view, hidden from this angle. The soft glow of the fireplace on the far side of the room adds warmth to the scene.

- Looking up at the ceiling, the room features modern recessed lighting, casting a soft, even glow across the space. Wooden beams accent the edges of the ceiling, adding a rustic touch. The tops of the windows and the moving curtains are visible from this angle, as the diffuse light from outside fills the room.

- Looking down at the floor, you see polished wooden floorboards and a stylish area rug that lies under the coffee table. The legs of the sofa are visible at the edge of the rug, and a few books and a small potted plant sit on the coffee table. The contrast between the rug and the wooden floor gives the room a balanced, warm feel.

**Snow landscape**

- A view of a frozen river winding through a snowy forest. The river is partially frozen, with snow-covered banks on either side and dark, still water visible in the center. Bare, leafless trees line the edges of the river, their branches covered in snow. The overcast sky casts a soft, cold light over the scene, creating a peaceful and serene winter atmosphere.

- A side view of the snowy forest, showing the frozen river cutting through the landscape. Bare trees with snow-covered branches stretch across the scene, and the forest extends into the distance, with the winding river creating a natural divide. The snowy ground and trees give the area a quiet, isolated feel, and the soft light from the sky casts long shadows across the snow.

- A view from behind the river, where the water flows into the distance, disappearing into the snow-covered forest. The winding shape of the river is prominent, with the snow forming smooth, white edges along the banks. The bare trees on either side create a tunnel-like effect as they stretch over the river. The air feels crisp and cold, and the entire landscape is blanketed in a thick layer of snow.

- A left side view of the river, with snow-covered branches hanging over the water. The frozen river winds through the snowy forest, with the bare trees standing tall on either side. The snow is thick and untouched, creating a pristine winter scene. The soft light of the overcast sky adds a calm, cold atmosphere to the landscape.

- An aerial view of the frozen river, showing its winding path through the snow-covered forest. The dark water contrasts sharply with the white snow, and the leafless trees form a web of branches stretching across the landscape. The river curves gently through the scene, and the blanket of snow gives the forest a peaceful, quiet feel.

- Looking down at the frozen river from above, the snow forms a thick blanket along the banks, with patches of dark water visible in the center. The ground is covered in snow, with bare tree branches reaching over the river. The scene feels still and cold, with no signs of movement, and the snow seems to absorb all sound, creating a peaceful, quiet atmosphere.

For completeness, we provide the text prompts used for the qualitative comparison on the Laval Indoor dataset. On the left, we depict the input image and on the right be provide the text prompt.

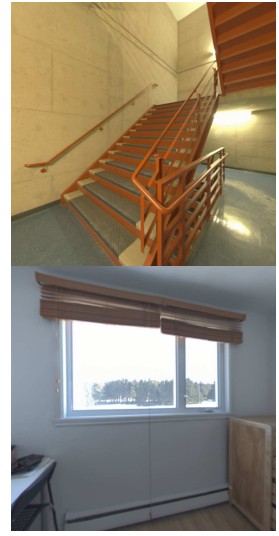

A concrete stairwell with orange railings leads up
to a yellow door with a number 2 on it.

A bedroom with a window overlooking a snowy
forest, a bed, a desk, and a dresser.

## A.3    PERSPECTIVE IMAGES FOR EVALUATION

Please note that all perceptual/text alignment metrics require another network to be computed. However, as these networks are trained with perspective images alone, the metrics would not give meaningful results when computed on panoramas. To circumvent this problem, we instead we render 10 random perspective images with a FOV of 90° for each panorama and use those for metric computation. Notice that we do not sample with an elevation of less than -45° or more then 45° as other works such as MVDiffusion do not generate full 360° panoramas.

## A.4    USER STUDY

As described in the main paper we perform a two-alternative forced choice (2AFC) considering all competitors and variants of our method. In Figure 3, we show the percentage of wins against all 1258 draws and corresponding confidence intervals. This indicates ours methods performs significantly better in the user study.

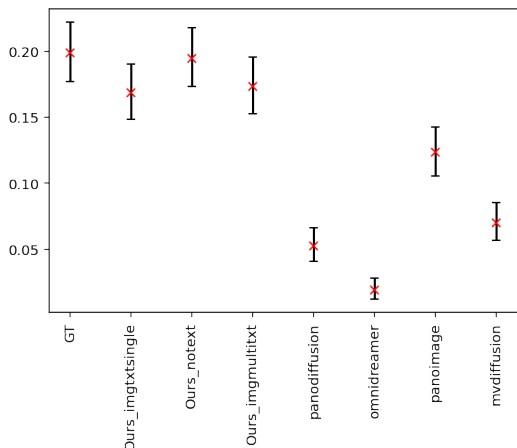

Figure 3: **Results of User study.** In this figure, we show the percentage of wins against all draws including the confidence interval

## A.5 MORE QUALITATIVE RESULTS

We provide more qualtitative results. In Figures 4 and 5, we show the input modalities in green and the generated faces and panorama. We see that our model aligns well to the given text prompts.

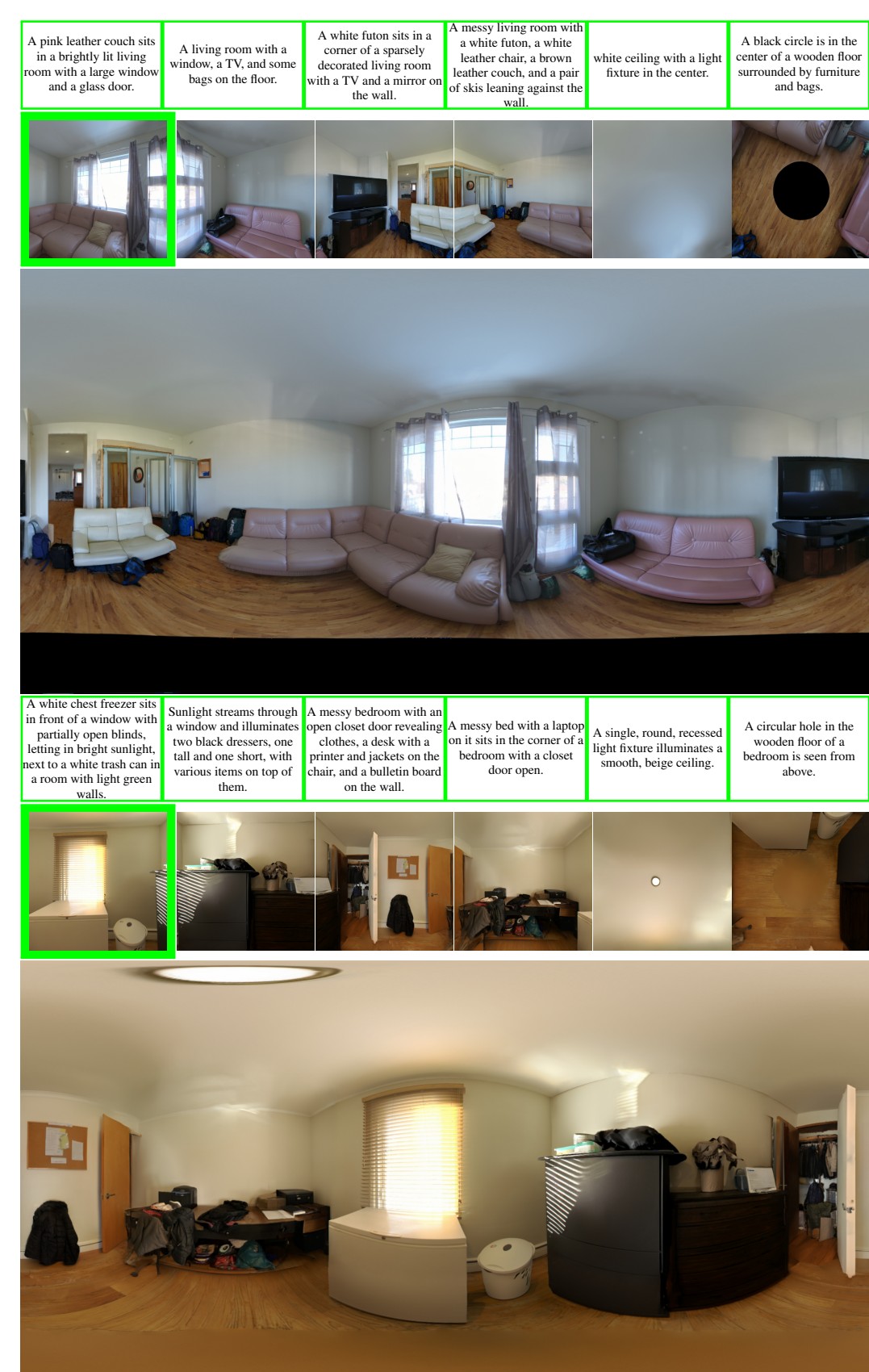

Figure 4: **Generated panoramas with multiple text prompts and image condition**

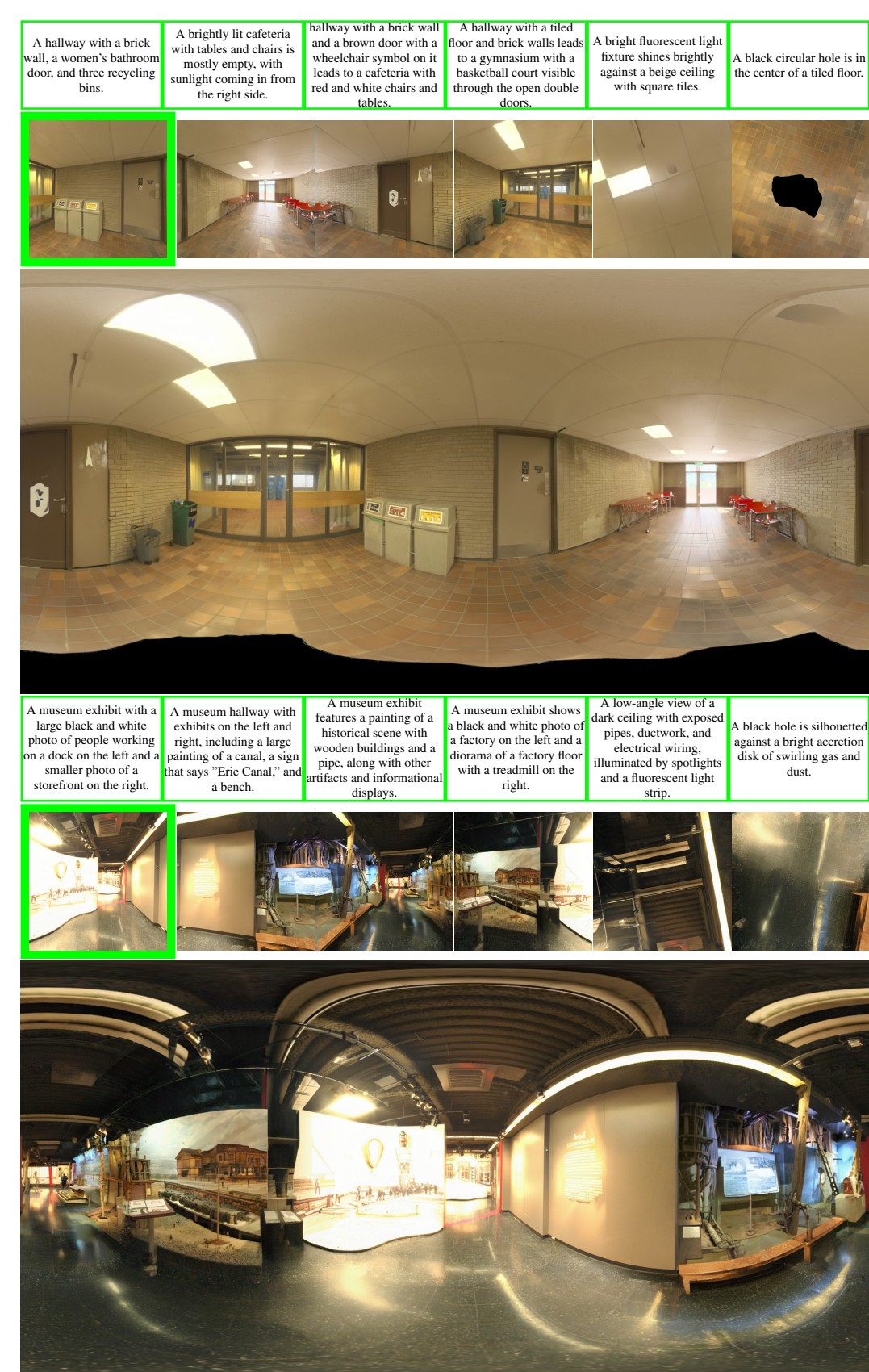

Figure 5: **Generated panoramas with multiple text prompts and image condition**

## A.6 MORE RESULTS OF UNSYNCHRONIZED GROUPNORM

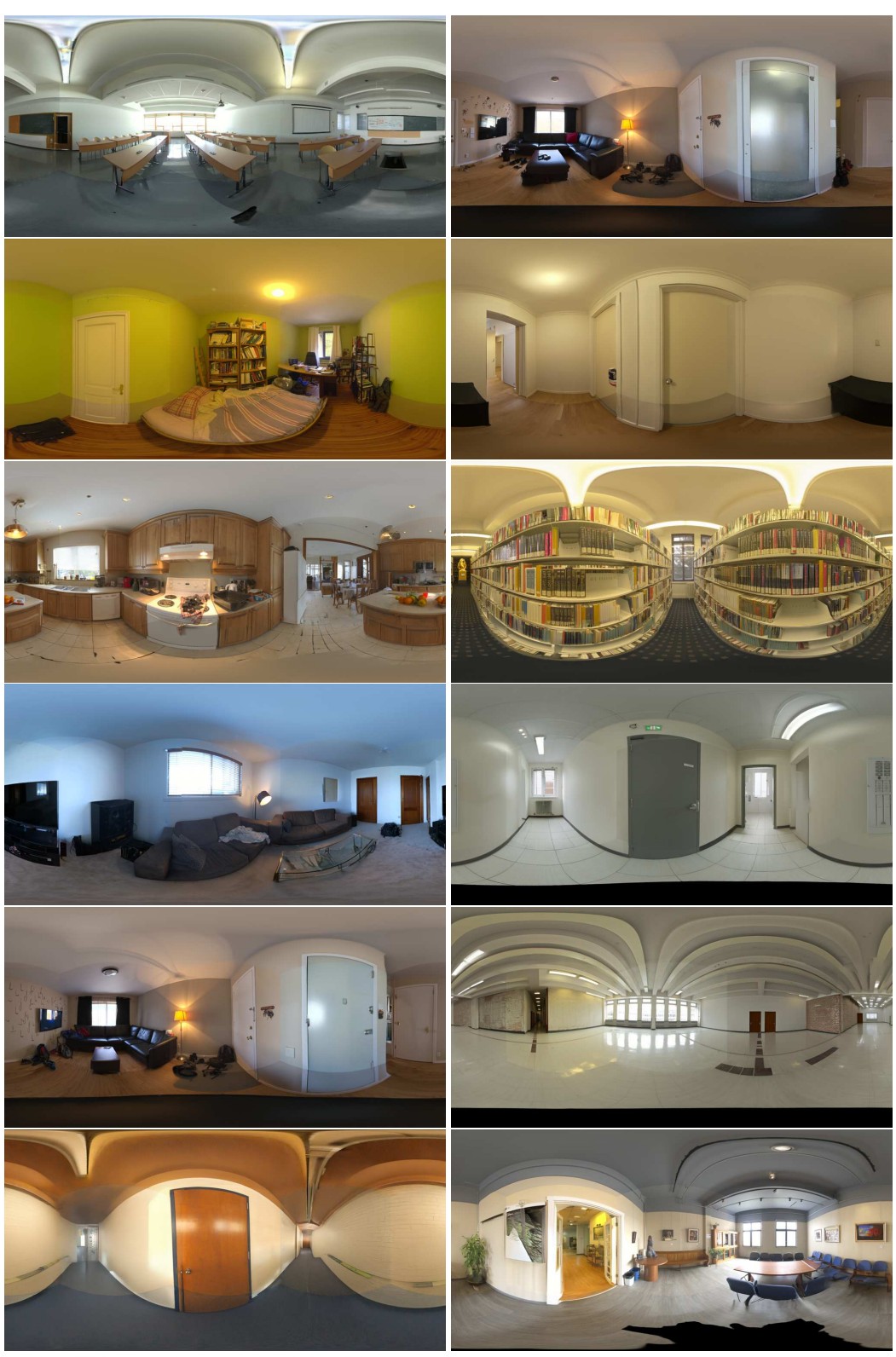

Figure 6: **Additional results of predictions with unsynchronized Group Norm.**

## A.7 MORE RESULTS OF NON-OVERLAPPING PREDICTIONS

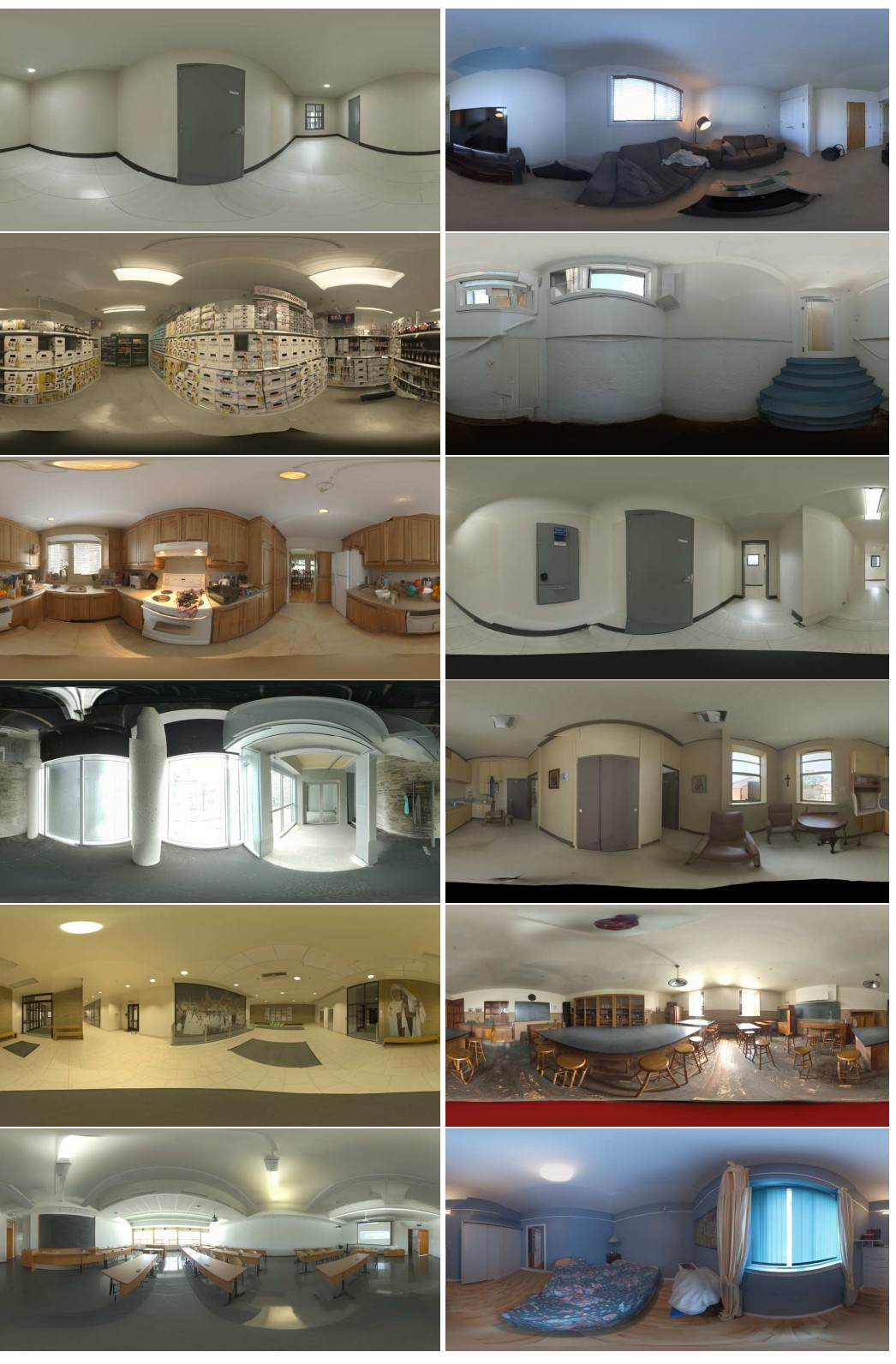

Figure 7: **Additional results of non-overlapping predictions.**

## A.8 Individual face overlaps from qualitative comparison

Here, we depict the individual faces generated by our three CubeDiff methods used in the equirect-angular panoramas in Figure 4 in the main paper. We show both uncropped and cropped faces as requested by reviewers. Additionally, we show the ground truth panoramas, the individual textual face descriptions. The corresponding input conditioning image is always the first (front) image of the individual faces (and thus equal for all models).

**Ours**$_{img}$

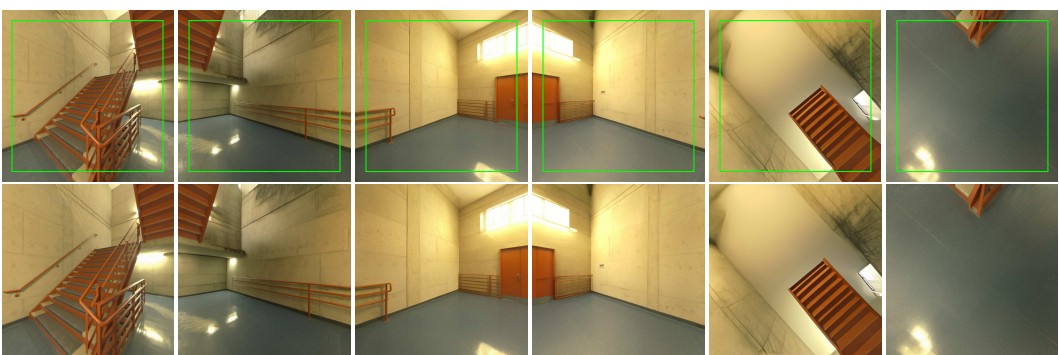

Figure 8: Our generated faces with input conditioning image only. Top row shows the uncropped faces, bottom row shows the cropped faces.

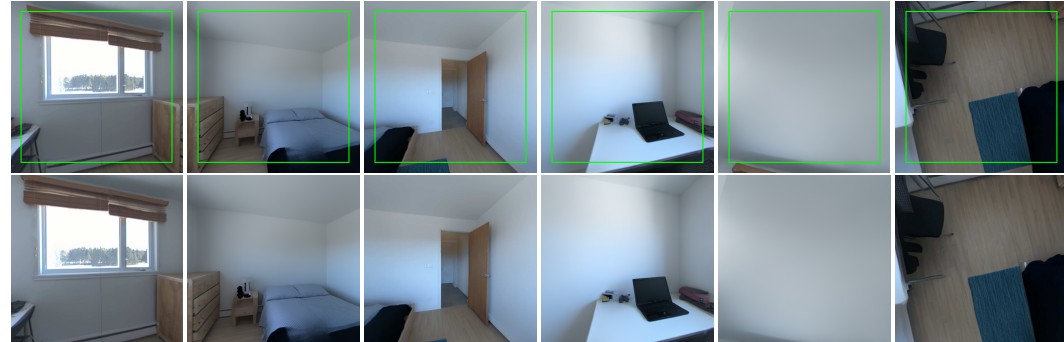

Figure 9: Our generated faces with input conditioning image. Top row shows the uncropped faces, bottom row shows the cropped faces.

**Oursimg+txt**

- A concrete stairwell with orange railings leads up to a yellow door with a number 2 on it.

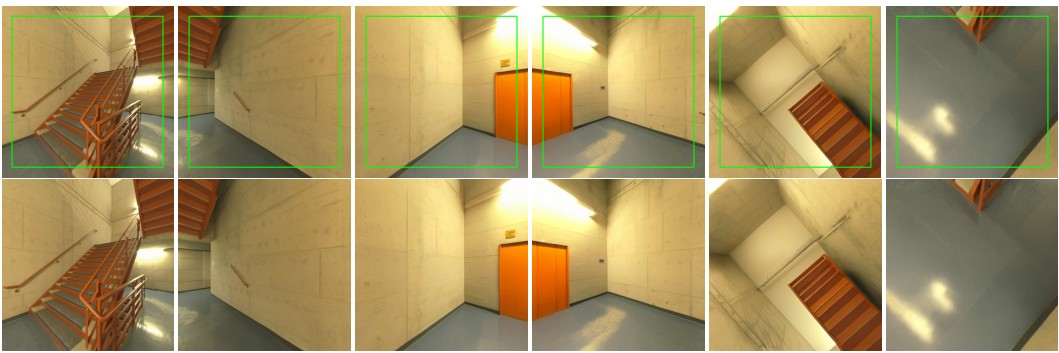

Figure 10: Our generated faces with single caption input. Top row shows the uncropped faces, bottom row shows the cropped faces.

- A bedroom with a window overlooking a snowy forest, a bed, a desk, and a dresser.

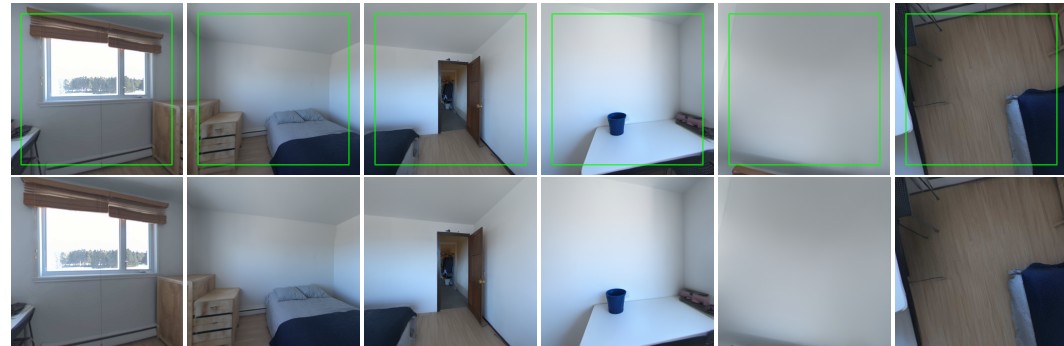

Figure 11: Our generated faces with single caption input. Top row shows the uncropped faces, bottom row shows the cropped faces.

**Ours**$_{\text{img+multitxt}}$

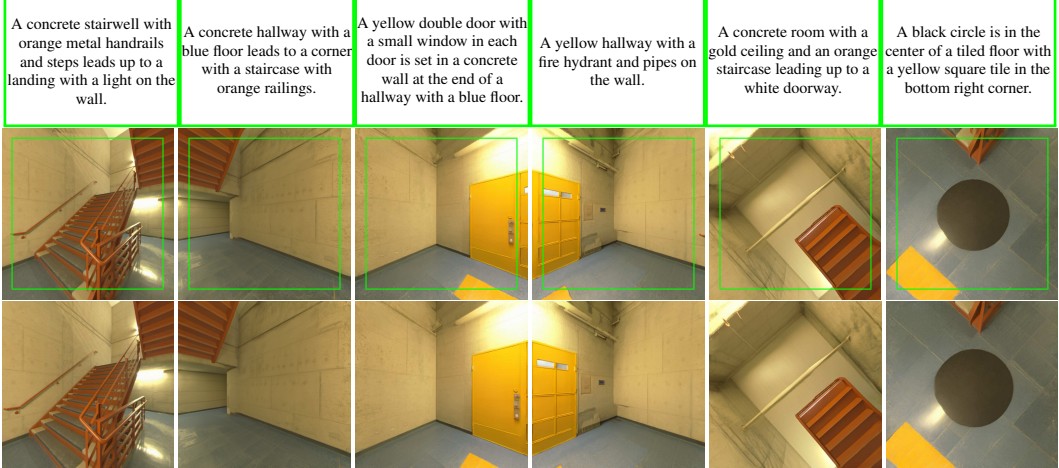

Figure 12: Our generated faces with multi caption input. Top row shows the uncropped faces, bottom row shows the cropped faces.

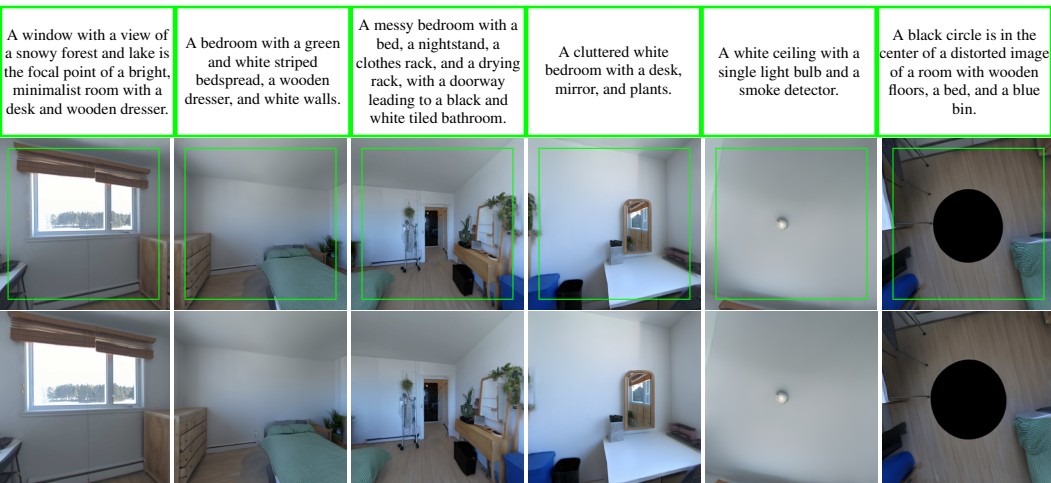

Figure 13: Our generated faces with multi caption input. Top row shows the uncropped faces, bottom row shows the cropped faces.

**Ground truth**

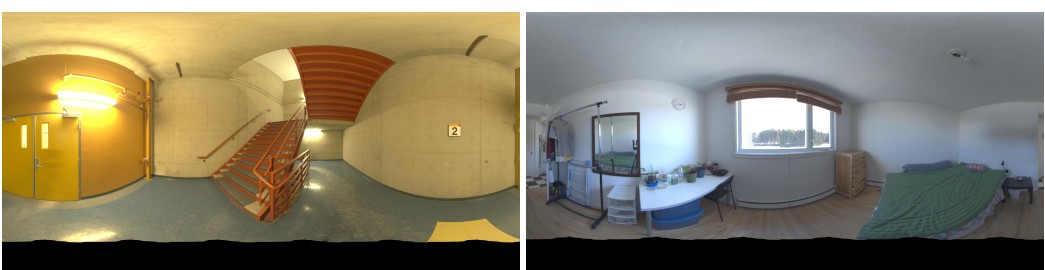

Figure 14: Ground truth panoramas from the Laval Indoor dataset.

## A.9 VAE RECONSTRUCTIONS

Below, we present pairs of images: the ground truth perspective images with a 95° field of view (FoV) and their corresponding reconstructed images. The reconstructed images are produced by passing the ground truth images through the encoder of our VAE and then decoding the resulting latent representations using the decoder of the same VAE.

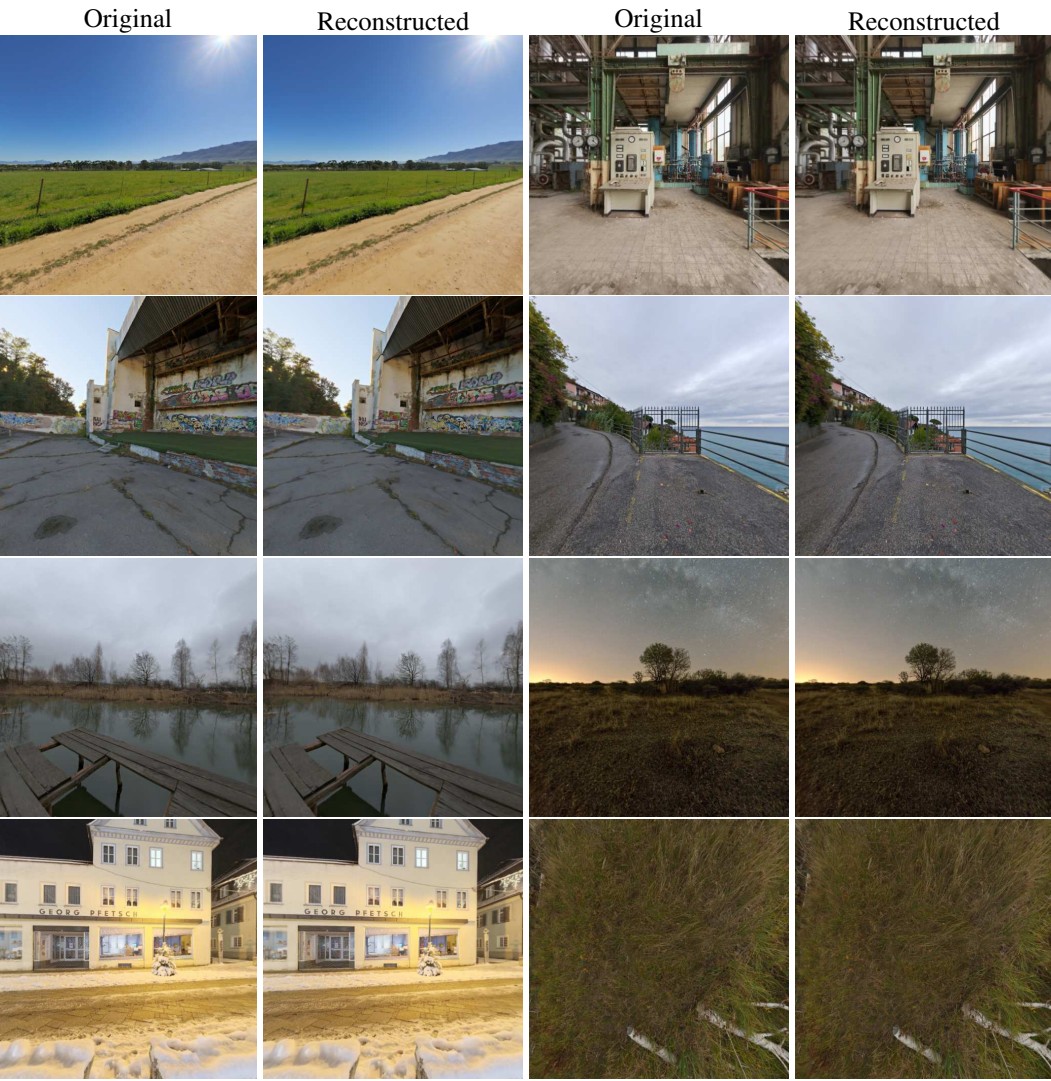

Figure 15: Examples of ground truth and encoded-decoded perspective images with a FoV of 95° using our VAE. The VAE is capable of reconstructing perspective images with out loss of quality.

## A.10 DETAILED ARCHITECTURE

We illustrate our latent diffusion model in Figure 16. The model's input is a concatenation of encoded latents, positional encodings, and an input mask indicating the conditioning image. To generate the initial latents, the input image is encoded with a VAE, while Gaussian noise is sampled for the other five faces. The VAE architecture is identical to that of Stable Diffusion's VAE, with one modification: all GroupNorm layers are replaced with synchronized GroupNorms, where normalization is computed across both the spatial and frame dimensions.

The combined input is downsampled three times to a resolution of $B \times 6 \times 32 \times 32$. The first and last blocks of the model exclude attention layers and operate independently on each face. Once the final layer is computed, the output is processed through the synchronized decoder. Notably, except for the GroupNorm layers, all computations are performed per face, with no awareness of the overall panorama structure.

Despite this simplicity, our approach outperforms existing methods, as demonstrated by our results.

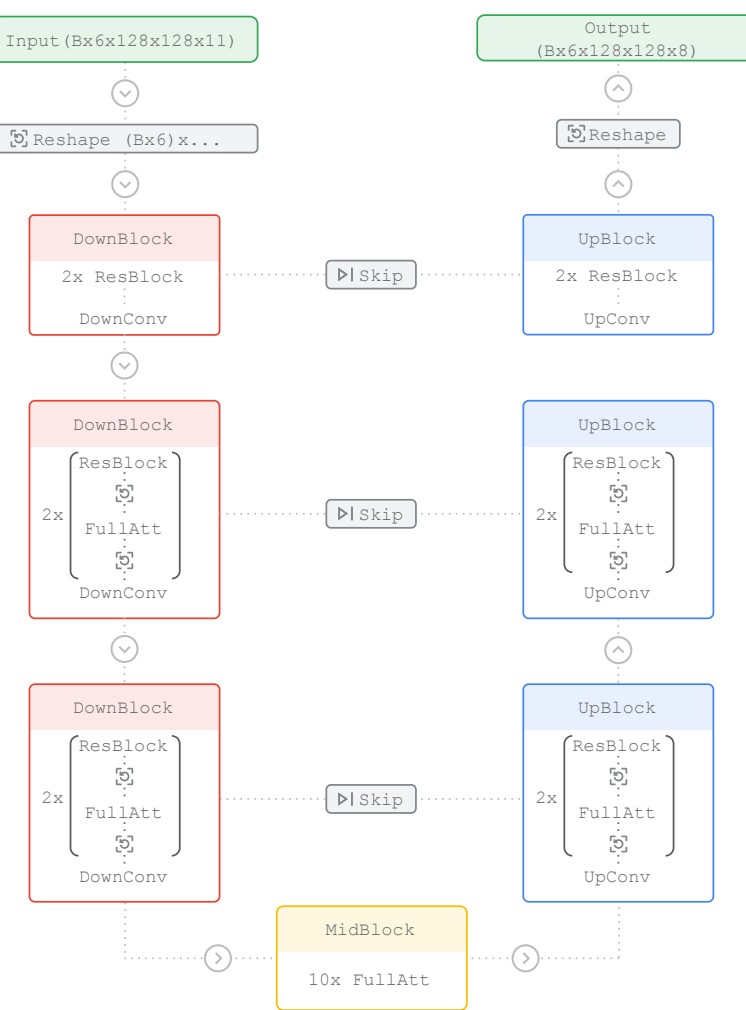

Figure 16: Illustration of our latent diffusion model.

## A.11 ABLATIONS ON PANORAMIC DATA

To evaluate the impact of dataset size on the performance of our method, we conducted an ablation study by training CubeDiff on three subsets of panoramic data: a tiny dataset containing approximately 700 panoramas from the Polyhaven dataset, a medium dataset of about 20,000 panoramas from the Structured3D dataset (the same dataset PanoDiffusion used and comparable in size to MVDiffusion), and a full dataset with over 40,000 panoramas. The results demonstrate that Cube-Diff performs robustly across all settings. Even the tiny model, trained on only 700 panoramas, achieves competitive results, while the medium model closely matches the performance of the full model and significantly outperforms baseline methods in most metrics. Qualitative results further confirm the ability of the tiny and medium models to generate visually consistent and high-quality panoramas, demonstrating CubeDiff's robustness even with constrained data. These findings indicate that the superior performance of CubeDiff stems not only from data volume but also from the strength of the cubemap representation and its compatibility with pretrained latent diffusion models.

| | LAVAL Indoor | | | SUN360 | | |
|---|---|---|---|---|---|---|
| | FID ↓ | KID ($\times 10^2$)↓ | Clip-FID ↓ | FID↓ | KID ($\times 10^2$)↓ | Clip-FID↓ |
| Text2Light | 28.3 | 1.45 | 11.5 | 60.1 | 4.31 | 31.3 |
| PanFusion | 41.7 | 2.85 | 19.8 | 30.0 | 1.42 | 7.8 |
| OmniDreamer | 71.0 | 5.17 | 23.9 | 92.3 | 8.89 | 51.7 |
| PanoDiffusion | 58.6 | 4.08 | 26.6 | 52.9 | 3.51 | 28.9 |
| Diffusion360 | 33.1 | 2.07 | 16.9 | 45.4 | 3.73 | 18.5 |
| MVDiffusion | 25.7 | 1.11 | 13.5 | 50.9 | 3.71 | 15.4 |
| **Ours**$_{tiny}$ | 27.3 | 1.05 | 8.8 | 41.7 | 2.99 | 14.7 |
| **Ours**$_{medium}$ | 13.8 | 0.66 | 8.5 | **23.9** | **1.28** | 10.7 |
| **Ours**$_{full}$ | **10.0** | **0.35** | **4.1** | 24.1 | 1.33 | **7.0** |

Table 1: **Quantitative Ablation on the Laval Indoor and SUN360 dataset.** We train a model (Ours$_{tiny}$) on a tiny dataset and another model (Ours$_{medium}$) on a medium dataset

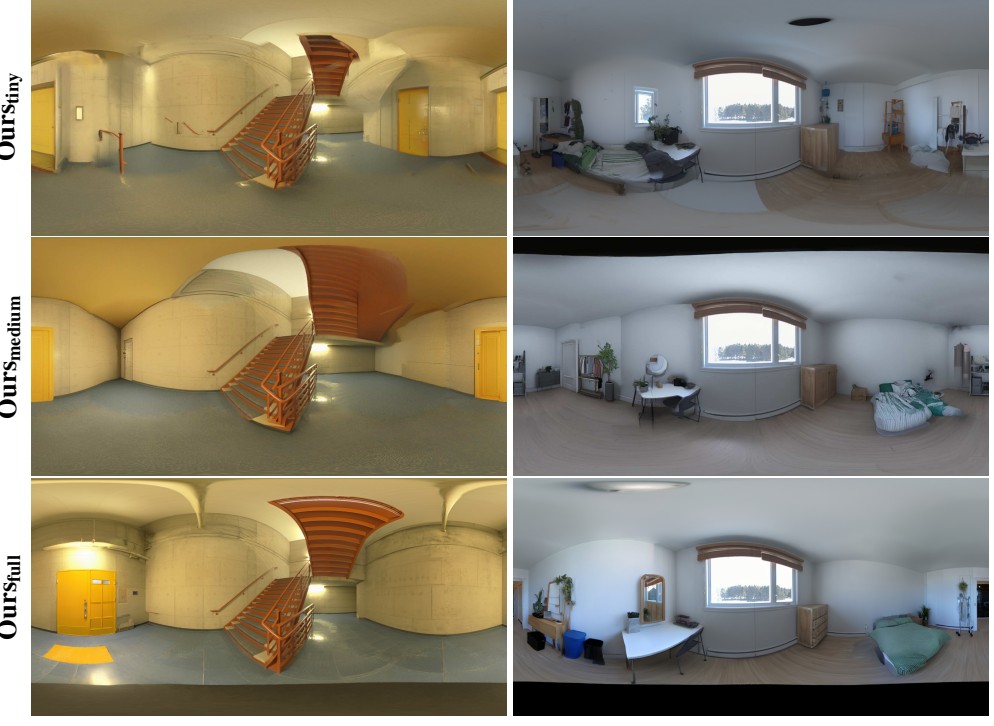

Figure 17: Qualitative results of the ablated models. Both the tiny and the medium model are able to generate consistent panoramas.

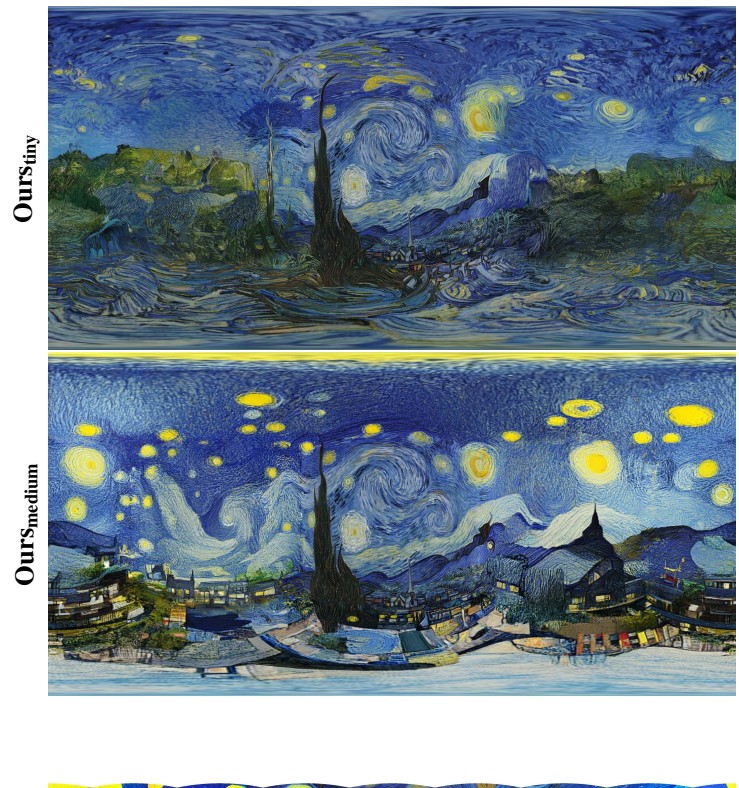

Figure 18: Example of an OOD generation of our tiny (top row) model, medium model (second row) and MVDiffusion (last row).