# OpenReview forum: "CubeDiff: Repurposing Diffusion-Based Image Models for Panorama Generation"
_ICLR.cc/2025/Conference — ICLR 2025 Spotlight_

### Official Review · Reviewer_FHAm · 2024-10-29

**Soundness:** 3
**Presentation:** 4
**Contribution:** 3
**Rating:** 8
**Confidence:** 4

**Summary:**

This paper presents a thoughtful approach to cubemap generation. The authors’ methodology is straightforward yet effective, addressing important challenges in cubemap generation, and achieves outstanding results compared to state-of-the-art.

**Strengths:**

This is a solid, simple yet effective work.

First, desipte the idea of using cubemap generation for panorama is not new,  the authors' solution through cubemap mv diffusion is simple but effective. This work offers an effective solution through overlapping predictions to address the wellknown issue of cubemap discontinuity.

Second, since each frame is generated individually (even with shared attention), tone shifts can occur across views. The authors address this by incorporating GroupNorm in VAE, which is a great application of existing techniques to solve this specific issue.

Lastly, their solution, while simple, performs significantly better than other alternatives, demonstrating that careful adjustments can yield meaningful improvements.

**Weaknesses:**

1. Lack of deeper analysis on how discontinuity is addressed: The proposed approach of overlapping predictions to handle discontinuity is interesting, but its effectiveness is not entirely clear. Overlapping predictions will bring new issues:  discontinuities within each view due to the six-view projection.  I recommend a more in-depth exploration:
- Visualization of the overlapping predictions: How does each view appear before panorama merging?
- Techniques for managing discontinuity within each view. Now each view might have severe boundary inside due to the discontinuity. Does the model handle this fine?  Is the achieved primarily due to VAE finetuning? How is shared attention helping here? Maybe attention visualization will benefit analysis.

2. No ablation study on component effectiveness: An ablation study would strengthen the paper, particularly with results on:
- The impact of GroupNorm
- The impact of overlapping prediction

**Questions:**

- Section 4.5 could be removed by briefly mentioning these details in the implementation section.
- The implementation details could benefit from more specifics, such as the base model used.
- Additional information on the VAE and UNet architectures would be helpful, particularly regarding the base models and the placement of GroupNorm in the VAE.
- Why did the authors choose to concatenate positional encodings as additional channels? It’s more common to add positional embeddings directly to the latent representation along with timestep embeddings.
- Is this work the first one to address panorama generation through cubemap diffusion?

---

> ### Author Response · Authors · 2024-11-22
> **Response to Reviewer FHAm**
>
> We thank Reviewer FHAm for the detailed review. In the following, we try to assess all questions and concerns.
>
> **Regarding the first point**:
>
> Thank you for requesting a deeper analysis of how discontinuities are addressed. This makes sense. We had in fact visualized the overlapping predictions in different views in Figure 4 (e.g., the duplicated fireplace in the right and back views). But we agree this should be made clearer in the text, and the illustrations. We also want to refer to the appendix, where the faces are visualized in Figure 10 and 11. The overlaps are clearly visible there and illustrate how predictions are managed in these regions.
>
> Discontinuities at the boundaries of the overlap do not matter, these parts are discarded. The important point is that the pixels within the overlap area are in perfect agreement, such that cropping yields seamless transitions between adjacent cube faces. In our experiments we find that the model reliably predicts these overlaps as it should, we have not observed any issues with discontinuities.
>
> To clarify, the VAE is not fine-tuned at all, it remains frozen throughout. It is exclusively the diffusion loop in latent space that is responsible for generating correct overlaps. Full attention is critical in this context, as it captures global contextual relations between pixels in different views, ensuring consistency.
>
> **Regarding the second point**:
>
> You are right, we should have included those ablations and have now added them in the main paper and the appendix. In particular, we qualitatively demonstrate the impact of GroupNorm and of overlapping predictions (quantitative performance metrics are not sensitive to the visually obvious differences, see detailed explanation in our response to Reviewer 1).

---

> > ### Comment · Reviewer_FHAm · 2024-11-25
> > **Reply the reponse**
> >
> > Thank you for your response.
> >
> > Regarding the overlapping predictions, I would like to see the following examples (referring to Figure 4):
> > - The original six predicted views (without cropping the overlapping regions). I want to understand how the original predictions look.
> > - The six views after cropping.
> > - The ground truth for the six views. I see some distortion in Fig4, it is better to include what they should look like in ground truth.
> >
> > For the VAE, if the VAE is frozen, how does it handle the original six views? I suspect the input views might be out of the distribution of the training images used for the VAE. I would like to know:
> > - What do the input and VAE reconstructions look like for the overlapped cubemap images.
> >
> > Finally, you seem to have addressed only the weaknesses while ignoring my questions part.

---

> > > ### Author Response · Authors · 2024-11-26
> > > **Response to Reviewer FHAm**
> > >
> > > > Regarding the overlapping predictions, I would like to see the following examples (referring to Figure 4):
> > > > * The original six predicted views (without cropping the overlapping regions). I want to understand how the original predictions look.
> > > > * The six views after cropping.
> > > > * The ground truth for the six views. I see some distortion in Fig4, it is better to include what they should look like in ground truth.
> > >
> > > Thank you for your feedback. In response, we have added an additional section in the rebuttal that visualizes the overlapping predictions referenced in Figure 4. This section includes:
> > >
> > > 1. Original Uncropped Predictions: The first row shows the original, unaltered predictions without cropping the overlapping regions for each of our models.
> > > 2. Cropped Views: The second row presents the six views after cropping the overlapping areas.
> > > 3. Ground Truth Panoramas: We include the ground truth panoramas from the Laval Indoor dataset for comparison. Note that only the center portion (the front image) of these panoramas serves as the input for our models, and this input is always depicted as the first image in each row.
> > >
> > > Additionally, while we can provide perspective images of these panoramas upon request, we believe this may be misleading since these perspective views are not used during inference or evaluation. We hope this additional section addresses your concerns and provides the clarity you were looking for.
> > >
> > > > For the VAE, if the VAE is frozen, how does it handle the original six views? I suspect the input views might be out of the distribution of the training images used for the VAE. I would like to know:
> > > > * What do the input and VAE reconstructions look like for the overlapped cubemap images.
> > >
> > > Thank you for your comment. The VAE in our approach is trained for general image reconstruction and incorporates a diverse set of perspective images, including those with a 95° field of view. This range aligns with many in-the-wild images, ensuring that the input views are not out of distribution.
> > >
> > > To address your suggestion, we have added an additional section in the appendix that showcases pairs of ground truth and corresponding VAE-reconstructed images for the overlapping cubemap views. As these results demonstrate, there is no visual degradation in the reconstructions, confirming that the VAE effectively handles these perspective images.
> > >
> > > Additionally, all generated panoramas presented in the paper inherently validate the capability of the frozen VAE to generate high-quality 95° perspective images, further supporting its robustness. We hope this clarification and the new section in the appendix address your concern.
> > >
> > > > Finally, you seem to have addressed only the weaknesses while ignoring my questions part.
> > >
> > > Thank you for pointing out that we missed all your questions. We focused on answering all weaknesses and forgot to answer your questions. Below, we try to address all your questions in detail.
> > >
> > > > * Section 4.5 could be removed by briefly mentioning these details in the implementation section.
> > >
> > > We agree that some parts of Section 4.5 overlap with the implementation details. However, we believe it is valuable to keep this section. We believe that classifier free guidance is interesting in our case, as we have it implemented both for text and for the input image - a less common design choice.
> > >
> > > > * The implementation details could benefit from more specifics, such as the base model used.
> > > Additional information on the VAE and UNet architectures would be helpful, particularly regarding the base models and the placement of GroupNorm in the VAE.
> > >
> > > We appreciate the concerns raised and included all the necessary details for reproducing CubeDiff with an open-source diffusion model in the appendix, in particular the UNet architecture. Regarding the VAE, we use the standard Stable Diffusion VAE with the only difference that we synchronize all GroupNorms. The code for SD's VAE is publicly available. More details in the answer below.
> > >
> > >
> > > > * Why did the authors choose to concatenate positional encodings as additional channels? It’s more common to add positional embeddings directly to the latent representation along with timestep embeddings.
> > >
> > > We chose to concatenate positional encodings directly to the latent representation because our positional encoding can naturally be represented as an image projected from the sphere to the image plane. This makes concatenation to the "image" latent representation more intuitive and effective. This approach is similar to existing methods, such as Cat3D [1], where positional encodings are directly concatenated to latent representations. In contrast, for text embeddings, positional encodings are typically combined with the timestep embeddings.

---

> ### Comment · Reviewer_FHAm · 2024-11-25
> **Response Reply part 2**
>
> 1. Fig. 7 in the main paper for the added ablation study is good. Could you merge Fig. 6 and Fig. 7 in the appendix to compare Non-Sync, No-overlap, Yours, and Ground Truth in each row? It is hard to interpret without putting them together.
>
> 2. I lowered my score because the rebuttal fails to address my concerns:
>    - Overlap prediction. How does the prediction look in its original uncropped version? How does the VAE handle it (e.g., reconstruction quality)? How does the prediction compare to the ground truth?
>
>    - Novelty.  Is this work the first to use cubemap for panorama generation? I noticed you cite RoomDreamer, which introduces cubemap for 3D synthesis as well. Please provide a full history of using cubemap for panoramas and compare your work to theirs in the related work section.
>
>    - Reproducibility. Do you use any pretrained LDM? In Line 192, you mentioned adopting an architecture similar to SD, so I assume you introduced a new architecture and trained it from scratch. Why not build upon SD and use the pretrained weights as much as possible? Please also include other important implementation details that might help reproduce your work.

---

> > ### Author Response · Authors · 2024-11-26
> > **Response to Reviewer FHAm - part 2**
> >
> > > * Is this work the first one to address panorama generation through cubemap diffusion?
> > > * Novelty. Is this work the first to use cubemap for panorama generation? I noticed you cite RoomDreamer, which introduces cubemap for 3D synthesis as well. Please provide a full history of using cubemap for panoramas and compare your work to theirs in the related work section.
> >
> > We are happy to answer your questions regarding the history of cubemaps for panorama generation. To the best of our knowledge, we are the first to use this representation directly to generate full 360° x 180° panoramas using a multi-view diffusion approach. Existing work mostly uses cubemaps as a surrogate loss or module to generate panoramas.
> >
> > * MVDiffusion [2] is the only one that also directly trains in perspective space but generates only eight perspective images, missing the top and bottom views. As a result, the output is incomplete and does not fully cover 360° × 180°. Additionally, their implementation requires training network components from scratch, while we can fully utilize pretrained latent diffusion models.
> > * RoomDreamer [3] addresses the challenge of synthesizing both geometry and texture aligned to the input scene structure and prompt simultaneously, i.e. it requires an existing mesh of a room and only stylizes the room with a new texture. In particular, their model receives as input six existing texture views and depth views, while ours only gets a single view and generates the other views from scratch. Importantly, it assumes the availability of an existing room mesh and primarily stylizes the room but cannot generate completely new panoramas or generalize to out-of-distribution environments, such as outdoor scenes.
> > * Like RoomDreamer, DreamSpace [4] relies on existing room meshes for cubemap texturing. Their findings suggest that equirectangular diffusion models perform better for texturing existing rooms, which contrasts with our results.
> > * Opa-Ma [5] introduces an outpainting approach, i.e. it generates panoramas iteratively. At every step, their diffusion model is guided by a near field of view condition and an omni-aware condition. The former is used for inpainting and updating the current panorama and the latter is used to guide overall consistency. For the omni-aware condition they use cubemaps, which means until the last step the cubemap contains all empty regions as well.
> > * DiffPano [6] method directly generates equirectangular panoramas using a diffusion model but uses cubemaps only for creating text-panorama pairs rather than directly for panorama generation.
> > * DiffCollage [7] is a general theoretical framework for diffusion models, similar to MultiDiffusion [8]. It reformulates the diffusion objective as a factor graph to generate large-scale content. The approach is completely different from ours, but they show that they can formulate cubemaps as factor graphs. Since the diffusion model only diffuses the same content in the overlapping parts, the method suffers from consistent geometry generation. They need segmentation masks to ensure logical scene structures, such as ensuring there is only one bed in a bedroom or one sun in the sky.
> >
> > We hope this extended perspective clarifies the use of panoramas in literature.
> >
> > [1] Gao, Ruiqi, et al. "Cat3d: Create anything in 3d with multi-view diffusion models." arXiv preprint arXiv:2405.10314 (2024).
> >
> > [2] Tang, Shitao et al. “MVDiffusion: Enabling Holistic Multi-view Image Generation with Correspondence-Aware Diffusion.” ArXiv abs/2307.01097 (2023): n. pag.
> >
> > [3] Song, Liangchen, et al. "Roomdreamer: Text-driven 3d indoor scene synthesis with coherent geometry and texture." arXiv preprint arXiv:2305.11337 (2023).
> >
> > [4] Yang, Bangbang, et al. "Dreamspace: Dreaming your room space with text-driven panoramic texture propagation." 2024 IEEE Conference Virtual Reality and 3D User Interfaces (VR). IEEE, 2024.
> >
> > [5] Gao, Penglei, et al. "OPa-Ma: Text Guided Mamba for 360-degree Image Out-painting." arXiv preprint arXiv:2407.10923 (2024).
> >
> > [6] Ye, Weicai, et al. "DiffPano: Scalable and Consistent Text to Panorama Generation with Spherical Epipolar-Aware Diffusion." arXiv preprint arXiv:2410.24203 (2024).
> >
> > [7] Zhang, Qinsheng, et al. "Diffcollage: Parallel generation of large content with diffusion models." 2023 IEEE/CVF Conference on Computer Vision and Pattern Recognition (CVPR). IEEE, 2023.
> >
> > [8] Bar-Tal, Omer, et al. "Multidiffusion: Fusing diffusion paths for controlled image generation." (2023).

---

> > ### Author Response · Authors · 2024-11-26
> > **Response to Reviewer FHAm - part 3**
> >
> > > Do you use any pretrained LDM? In Line 192, you mentioned adopting an architecture similar to SD, so I assume you introduced a new architecture and trained it from scratch. Why not build upon SD and use the pretrained weights as much as possible? Please also include other important implementation details that might help reproduce your work.
> >
> > Our latent diffusion model (LDM) is indeed pretrained, which is a critical component of our approach. One of the core strengths of using cubemaps is their compatibility with pretrained models, as each cubemap face falls within the range of perspective images typically used to train such powerful diffusion models. This alignment allows us to leverage the generalization capabilities of the pretrained model without the need to train from scratch. A compelling example of this is how our model successfully generalizes to paintings and extends them into complete panoramas. These painting-based examples are not part of our panorama training data, yet our model achieves this seamlessly, demonstrating the strength of both the pretrained LDM and the cubemap representation.
> >
> >
> > We hope that this response has addressed all of your questions. If there’s anything we may have missed or if you have additional questions, please don’t hesitate to let us know. We’d be more than happy to provide further clarifications or details!

---

> > > ### Comment · Reviewer_FHAm · 2024-11-27
> > > **Reply**
> > >
> > > Thank you for the detailed response.
> > > The history part is well-written, and a more concise version could be integrated into the paper.
> > >
> > > My remaining concern is the visualization of the original, uncropped predictions, as the current PDF does not yet reflect the updated version of Figure 4.

---

> > > > ### Author Response · Authors · 2024-11-27
> > > > **Response to Reviewer FHAm**
> > > >
> > > > Dear Reviewer,
> > > >
> > > > Thank you for your kind feedback and for recognizing our efforts in addressing your questions. We’re glad you found the history section well-written and appreciate your suggestion to integrate a concise version into the paper.
> > > > Regarding your remaining concern, could you clarify what you mean by "the original, uncropped predictions"? In Appendix A.8, we have included both the original uncropped predictions and the cropped predictions for all individual faces and all our models. If something specific is missing, please let us know.
> > > >
> > > > We did not update Figure 4 in the main paper for two reasons: (1) Figure 4 focuses on comparing CubeDiff with panorama baselines, most of which (except MVDiffusion) do not generate individual perspective images, and (2) space limitations prevent us from including the uncropped and cropped predictions for all three models. We believe the appendix is the best place for these details but are happy to revise the text in the main paper to better point readers to the appendix.

---

> > > > > ### Comment · Reviewer_FHAm · 2024-11-27
> > > > > **Reply**
> > > > >
> > > > > I mean the top row in appendix figure 8. I did not suggest add the uncropped visualization in main paper. I think you pointed me to main paper fig.4 for this updated visualization.  Anyway, i am happy with appendix fig. 8.

---

> > > > > > ### Author Response · Authors · 2024-11-27
> > > > > > **Response to Reviewer FHAm**
> > > > > >
> > > > > > There was probably a confusion due to the figure numbers changing when we updated the appendix. Great to hear that you are happy with Figure 8. Let us know if there is anything else!
> > > > > >
> > > > > > Cheers
> > > > > > Authors

---

> > > > > > > ### Comment · Reviewer_FHAm · 2024-11-27
> > > > > > > **reply**
> > > > > > >
> > > > > > > maybe add a bounding box along the boundary in the top row where you will crop to make the figure clearer.

---

> > > > > > > > ### Author Response · Authors · 2024-11-28
> > > > > > > > **Reply**
> > > > > > > >
> > > > > > > > We've added the bounding boxes to the uncropped images.

---

### Official Review · Reviewer_mPic · 2024-11-01

**Soundness:** 3
**Presentation:** 3
**Contribution:** 3
**Rating:** 8
**Confidence:** 3

**Summary:**

The paper presents a new method for generating panoramic images. The key idea of the method is to utilize cubemap projections, instead of equirectangular projection. The method is relatively straightforward, but demonstrates good results.

**Strengths:**

- It is a simple to understand and implement method.
- Paper is well written and easy to follow.
- Results seem to be better than competitive methods.
- New application of generating panorama based on multiple text-prompts.

**Weaknesses:**

- The claim of "fully recycle pretrained text-to-image model, enabling generalization far beyond the limited training data" is not well supported by results. The convincing results here would be, if the paper shows that the method can generate panoramas that are clearly outside of the panoramic dataset distribution. For example, stylistic panoramas, e.g. cartoons, oil painting or some fantasy concepts, such as "Alice in wonderland", etc.

- Table. 1 is confusing. First of all there is no column $Ours_{text}$, while in Sec 4.5. It is said that models can operate in this mode. Second coloring in this Table is confusing. My suggestion would be to provide green/light green coloring within each modality: "Text-only", "Image-only", etc.

**Questions:**

--

**Details Of Ethics Concerns:**

--

---

> ### Author Response · Authors · 2024-11-22
> **Response to Reviewer mPic**
>
> We thank Reviewer mPic for the detailed review. In the following, we try to assess all questions and concerns.
>
> **Regarding the first point**:
>
> Our test data (e.g., all panoramas in Figs. 1 and Figure 8) was totally distinct from the training data, but we agree that we should have included examples that are more obviously out-of-domain. As requested by the reviewer, we have now included more exotic styles like a panoramic extension of Van Gogh's "Starry Night", and a panorama based on "Alice in Wonderland" in the appendix. Please refer to Figure 9. We thank the reviewer for the suggestion, we feel that it has strengthened the paper and are happy to include the samples in the main paper in an updated version.
>
> **Regarding the second point**:
>
> Thank you for the feedback. Regarding Tabe 1, there seems to be a  misunderstanding. In our work we use intermediate image conditioning, i.e., we first run a conventional text-to-image model to generate one (the "front") face of the cube map, then feed both the generated front face and the text prompt into the panorama generator. As such, both *Oursimg+txt* and *Oursimg+multitxt* effectively operate in text-only mode. That approach is consistent with previous work such as CAT3D.
>
> The color scheme served only to highlight the best and second-best scores across all columns, as often done for large tables in machine learning papers. However, we appreciate the suggestion for improved clarity and have updated Table 1 accordingly.

---

> > ### Comment · Reviewer_mPic · 2024-11-23
> >
> > Dear Authors,
> > Can I kindly ask you to add ood result to the project website, it is very hard to judge them in pdf?
> > Regards, R mPic

---

> > > ### Author Response · Authors · 2024-11-23
> > > **Response to Reviewer mPic**
> > >
> > > Dear Reviewer, thank you for the suggestion. We have added the generated panoramas from the paper (and more :-)) to the website. They can be found under the tab "Ours OOD". You are right, that the 3D viewer adds a valuable (and better) perspective on the images. We hope this provides clarity and highlights the quality of our method.

---

> > > > ### Comment · Reviewer_mPic · 2024-11-25
> > > >
> > > > Thank you, I will increase my rating to 8.

---

> ### Comment · Reviewer_FHAm · 2024-11-25
> **Appreciation**
>
> The stylistic panoramas look great!

---

### Official Review · Reviewer_oYCw · 2024-11-02

**Soundness:** 3
**Presentation:** 3
**Contribution:** 2
**Rating:** 6
**Confidence:** 4

**Summary:**

This paper proposes CubeDiff, a method that generates panoramas from text or single-image condition. The core of CubeDiff is to represent a panorama with cubemaps, 6 images representing the 6 faces of a cube that seamlessly capture the scene in an inside-out way. With this representation, CubeDiff adapts a pre-trained text-to-image diffusion model to generate the 6 cube-face images simultaneously. Similar to previous works like MVDream and Wonder3D, the proposed method extends the self-attention layer in the pre-trained denoising network to operate on multiple images for learning holistic multi-view consistency. Experiments on Laval Indoor and Sun360 datasets show that CubeDiff outperforms previous panorama generation methods (MVDiffusion, Diffusion360, etc,) across various generation settings (text-conditioned, image-condition, or both).

**Strengths:**

1. CubeDiff's design choices are reasonable and backed up with a detailed discussion and comparison of different panorama representations. Extending the original self-attention over image patches of a single image to handle multiple views is a common strategy in the literature of multi-view diffusion models and is a natural choice here, considering the cube map representation.

2. The paper provides a detailed discussion on different representations for the panorama in Sec.3. Together with the related works (Sec.2), choosing cube maps as the generation targets of the multi-view diffusion model is well-motivated.

3. The experiments cover a rich set of conditioning settings, and CubeDiff demonstrates supreme performance both qualitatively and quantitatively. The anonymous website provides an interactive qualitative comparison, which helps better understand the quality of different approaches.

**Weaknesses:**

1. The experiment section of the paper lacks some critical ablation studies and is not complete: 1) Sec4.2 states the importance of the Synchronized GroupNorm, but there is no quantitative analysis for this design; 2) Sec4.4 claims the design of overlapping predictions between the nearby faces can reduce the generation artifacts, but again lack a corresponding ablation study.

2. The overall technical contribution is relatively limited - there are no new designs from the modeling perspective. However, given the good empirical performance and the simplicity of the approach, this is not necessarily a vital weakness. Therefore, I hope the authors can address the first point above to complete the ablation studies for the core modeling components described in Sec.3 and Sec.4.

3. It is unclear to me if the training data of the baselines (e.g., MVDiffusion and DIffusion360) is the same as the proposed CubeDiff. If they use different training data, would the comparisons be unfair? Ideally, the authors should re-train those diffusion-based baselines with the same training data.

**Questions:**

Please refer to the Weaknesses section for the questions.

In general, the proposed CudeDiff is simple and obtains good experimental results. However, the paper does not provide necessary ablation studies for some detailed design choices, and it's also unclear if the comparisons against baselines are conducted in a fair setting. These two points are especially important given CudeDiff's limited technical novelty. I will consider updating the rating if these concerns can be properly addressed.


-----

**After rebuttal**: I increased the score to 6 as the authors provided a thorough study with different amounts of training panoramas, which helps demonstrate the method's effectiveness and generalization capability.

---

> ### Author Response · Authors · 2024-11-22
> **Response to Reviewer oYCw**
>
> We thank Reviewer oYCw for the detailed review. In the following, we try to assess all questions and concerns.
>
> **Regarding the first point**:
>
> We agree that we should have better supported the decision to employ synchronized group normalization and overlapping predictions. We have now included qualitative results in the manuscript to demonstrate the impact of these components.
>
> Quantitative comparisons for these components are challenging: their main benefit is to prevent subtle color shifts and discontinuities at the border of cube faces, but existing metrics (including FAED) are not sensitive to these artifacts; whereas they stand out for human perception and markedly degrade visual quality. Hence, visual inspection offers a more meaningful insight into the effects of these components. More results can be found in the appendix.
>
> **Regarding the second point**:
>
> We concur that the pieces of the puzzle -- cube maps, full multi-frame attention, extending the token sequence length to keep extensively pretrained weights, synchronized group norm -- were all there. But we are adamant that the simplicity of our design is a strength rather than a weakness. Our goal was efficient and high-quality panorama generation, and we feel CubeDiff achieves that without cluttering the network diagram with arbitrary custom modules and branches. We are quite pleased that we found a way to assemble the right components into a clean architecture without bells and whistles, which, in hindsight, looks obvious.
>
> **Regarding the third point**:
>
> We respectfully disagree with the assertion that the comparisons are unfair. The goal was in-the-wild deployment, i.e., a zero-shot setting where the characteristics of the input image are not know in advance. All models, including CubeDiff as well as the baselines (except Diffusion360 and OmniDreamer), were evaluated in that same setting, i.e., on out-of-distribution data that none of them had been specifically trained for. In our view, this is an entirely fair comparison: the target is a method that works out of the box for most reasonable inputs, and the evaluation protocol emulates that setting. This form of evaluation is accepted standard practice in computer vision, and in machine learning at large.
>
> Besides the fact that the responsibility to choose suitable training data lies with the respective authors, re-training baselines on the same training set is not feasible, since some do not make their training code public. Retraining only a subset would, if anything, make the comparison more skewed.

---

> > ### Comment · Reviewer_oYCw · 2024-11-24
> >
> > Thank you for the response. However, it seems that it does not properly address any of my questions.
> >
> > 1. Several qualitative examples are not sufficient for rigorous ablation studies. If the authors believe none of the existing quantitative metrics can capture the improvement brought by those two designs, one alternative is to conduct a user study, which is quite common in evaluating image/video generation results.
> >
> > 2. As I have stated in the initial review, the limited novelty is not a vital weakness as long as the experiments can properly/sufficiently back up each design of the method. Unfortunately, given the latest response, I have more doubts about this.
> >
> > 3. The third point is a serious one. I'm pretty surprised to see this response. If you have much more training data than the baselines, how can you tell the source of your performance improvement (model or data?) -- “Control variables” should be the basics for scientific experiments. By checking those baseline papers, it looks like your training data is 3X more than the data of PanoDiffusion and 5X more than the data of MVDiffusion, so it is likely that previous methods can also achieve better results given the same amount of panorama training data. If the baseline methods do not provide training code,  you can at least try to train your model with different amounts of pano data to see how this affects the performance.

---

> > > ### Author Response · Authors · 2024-11-27
> > > **Response to Reviewer oYCw**
> > >
> > > > 1. Several qualitative examples are not sufficient for rigorous ablation studies. If the authors believe none of the existing quantitative metrics can capture the improvement brought by those two designs, one alternative is to conduct a user study, which is quite common in evaluating image/video generation results.
> > >
> > > We believe that the extensive qualitative results provided, particularly in the appendix, offer strong evidence supporting our design choices. These examples showcase the robustness and effectiveness of our method in various scenarios. However, we acknowledge the value of additional quantitative evaluation through a user study, which is a widely accepted approach for assessing image and video generation results. We are happy to provide a user study to quantitatively evaluate the impact of GroupNorm and overlap predictions in the final version of the paper.
> > >
> > >
> > > > 3. The third point is a serious one. I'm pretty surprised to see this response. If you have much more training data than the baselines, how can you tell the source of your performance improvement (model or data?) -- “Control variables” should be the basics for scientific experiments. By checking those baseline papers, it looks like your training data is 3X more than the data of PanoDiffusion and 5X more than the data of MVDiffusion, so it is likely that previous methods can also achieve better results given the same amount of panorama training data. If the baseline methods do not provide training code, you can at least try to train your model with different amounts of pano data to see how this affects the performance.
> > >
> > > We recognize the importance of ensuring a fair comparison between CubeDiff and baseline methods such as MVDiffusion and PanoDiffusion. While some baselines do not provide training code, we have taken significant steps to evaluate how the training data size influences the performance of CubeDiff. To address the concerns regarding the training data, we conducted an ablation study on dataset size. In addition to training our full model on over 40,000 panoramas, we trained two reduced variants:
> > >
> > > * A medium model trained on 20,000 panoramas (Structured3D dataset), the same as PanoDiffusion and similar to MVDiffusion’s 11,000 panoramas (both indoor datasets).
> > >
> > > * A tiny model trained on approximately 700 panoramas (Polyhaven dataset only), which is significantly smaller than any other method was trained on. For instance, 15 times smaller than MVDiffusion's training data.
> > >
> > > These results allow for a direct comparison under conditions similar to those of the baselines. We added quantitative and qualitative results of the ablation in the appendix. The quantitative results in our study demonstrate that our medium model achieves performance close to our full model and significantly outperforms baselines such as MVDiffusion, PanoDiffusion, and Diffusion360.
> > >
> > > Our tiny model (only 700 panoramas) generates visually appealing panoramas, outperforming several baselines in key metrics despite being trained on significantly less data than competitors. These results confirm that CubeDiff is robust and effective even under constrained data conditions. Qualitative examples further illustrate the ability of the tiny and medium models to produce coherent panoramas. We have also included an out-of-distribution (OOD) example comparing our two models with MVDiffusion. The results clearly demonstrate that both of our models generate panoramas that are significantly more visually coherent and appealing. These examples further support our arguments regarding the advantages of our approach in handling diverse and challenging scenarios.

---

> ### Comment · Reviewer_oYCw · 2024-11-25
> **Followup**
>
> Hi authors,
>
> I want to follow up on your updated OOD results in your anonymous project page: the conditioning image seems to be a panorama by itself rather than a perspective image, is it intentional or something went wrong there? Also, I wonder if you have further updates on the first and third points in our discussion.

---

> ### Comment · Reviewer_FHAm · 2024-11-25
> **Response to both Reviewer oYCw  and Authors**
>
> Hi all,
>
> Thanks for the really valuable discussions! First of all, I really like this work — it is very simple and seems highly effective. However, to be honest, I feel the authors seems failed to spend enough time on tackling some critical issues during rebuttal. That being said, I still vote for the acceptance of this work due to its simplicity and effectiveness.
>
> I fully agree with reviewer oYCw regarding the comparisons. First of all, the initial version lacks too many ablation studies. From the current version after the rebuttal, we still do not see the critical ablation study that demonstrates the strengths of this work and highlights the most important components of its contributions. There should be an ablation study the same architecture same training but without cubemap generation (which appears to be the most important component). Additionally, the authors should provide a baseline for an extremely simple version of the cubemap model without any other novel designs (e.g., groupnorm, overlap prediction, 3D attentions). This would serve as a baseline to show the importance of your other technical details and how it outperforms the simpler baseline.
>
> Regarding the training difference with MVDiffusion and other works, it makes sense. I feel adding ablation without cubemap generation can address this concern, and I feel this is the reason why reviewer asked this question: We do NOT know  the quality improvement comes from the most important component (Cubemap), or from the training difference. The ablations provided now are just minor things. I strongly these ablations during rebuttal or at least in camera ready if the paper accepted).
>
>
> Lastly, I really want to know the history of using cubemaps for panoramas, as I mentioned in my review but did not get an answer. This would help establish the novelty of your work.

---

> > ### Comment · Reviewer_oYCw · 2024-11-26
> > **Response to Reviewer FHAm**
> >
> > Hi Reviewer FHAm,
> >
> > Thank you for providing detailed thoughts!
> >
> > ```
> > I feel the authors seems failed to spend enough time on tackling some critical issues during rebuttal. That being said, I still vote for the acceptance of this work due to its simplicity and effectiveness.
> > ```
> > I completely agree. The proposed method delivers promising empirical and qualitative results. With properly designed ablation studies analyzing each component, this paper could become an impactful and inspiring work.
> >
> > ---
> >
> > ```
> > Regarding the training difference with MVDiffusion and other works, it makes sense. I feel adding ablation without cubemap generation can address this concern, and I feel this is the reason why reviewer asked this question: We do NOT know the quality improvement comes from the most important component (Cubemap), or from the training difference.
> > ```
> > Agreed. To address this concern, experiments with the following settings could be helpful:
> >
> > 1. Generate the entire equirectangular panoramas directly.
> > 2. Use the representation of MVDiffusion while retaining the cross-view self-attention mechanism used by CubeDiff.
> >
> > And the training setups must be the same for appropriate ablation studies.
> >
> > ----
> >
> > My initial review was intended to encourage the authors to conduct properly designed analyses, and I was open to increasing my score. Unfortunately, the rebuttal has been quite disappointing, as the authors did not seem to make a serious effort to address these concerns, and their response feels perfunctory. I will decrease my score if the authors do not actively engage in the discussion.

---

> > > ### Author Response · Authors · 2024-11-27
> > > **Response to Reviewer FHAm and oYCw**
> > >
> > > > Regarding the training difference with MVDiffusion and other works, it makes sense. I feel adding ablation without cubemap generation can address this concern, and I feel this is the reason why reviewer asked this question: We do NOT know the quality improvement comes from the most important component (Cubemap), or from the training difference.
> > >
> > > > Agreed. To address this concern, experiments with the following settings could be helpful:
> > > > Generate the entire equirectangular panoramas directly.
> > > > Use the representation of MVDiffusion while retaining the cross-view self-attention mechanism used by CubeDiff.
> > >
> > > We appreciate your suggestions and agree that exploring alternative settings can help to address your concerns. Regarding the first point, we can attempt to train a model that generates entire equirectangular panoramas directly using our latent diffusion model (LDM). While prior methods like 360Diffusion have shown limitations in this setting, we are open to exploring this approach to provide a direct comparison in the final version. However, we also want to highlight the inherent advantages of cubemaps, such as fine-grained text control and better handling of perspective distortions, which make them particularly effective for high-quality panorama generation.
> > >
> > > Regarding the second point, we believe that using the MVDiffusion representation while retaining CubeDiff’s cross-view self-attention mechanism is not a meaningful experiment. By design, MVDiffusion generates incomplete panoramas with missing top and bottom regions, and its smaller receptive field in attention layers inherently limits its ability to capture global context. These design choices are fundamentally different from CubeDiff, and we do not see how combining these approaches would yield useful insights or improve upon existing methods.

---

> > > > ### Comment · Reviewer_oYCw · 2024-11-27
> > > >
> > > > Thank you for the experiments with different data sizes. It's an important observation that your approach gets decent results with only 700 panoramas, which can strongly support your previous claims on preserving the generation priors of the pre-trained model.
> > > >
> > > > ```
> > > > We appreciate your suggestions and agree that exploring alternative settings can help to address your concerns. Regarding the first point, we can attempt to train a model that generates entire equirectangular panoramas directly using our latent diffusion model (LDM). While prior methods like 360Diffusion have shown limitations in this setting, we are open to exploring this approach to provide a direct comparison in the final version. However, we also want to highlight the inherent advantages of cubemaps, such as fine-grained text control and better handling of perspective distortions, which make them particularly effective for high-quality panorama generation.
> > > >
> > > > Regarding the second point, we believe that using the MVDiffusion representation while retaining CubeDiff’s cross-view self-attention mechanism is not a meaningful experiment. By design, MVDiffusion generates incomplete panoramas with missing top and bottom regions, and its smaller receptive field in attention layers inherently limits its ability to capture global context. These design choices are fundamentally different from CubeDiff, and we do not see how combining these approaches would yield useful insights or improve upon existing methods.
> > > > ```
> > > > I think there are still slight misunderstandings. These experimental settings are not for complete models/methods that lead to good performance. Instead, their goal is to help understand the importance of your core design (cubemap) under controlled experimental setups. In other words, they try to answer the question, "If you drop the cube map representation, how bad would the performance be?". I'm good with not having these experiments since you have thoroughly studied the impact of training data size, and my main concerns are resolved. I will increase my score accordingly.

---

> > > ### Author Response · Authors · 2024-11-27
> > > **Response to Reviewer oYCw and FHAm**
> > >
> > > > I feel the authors seems failed to spend enough time on tackling some critical issues during rebuttal.
> > >
> > > > My initial review was intended to encourage the authors to conduct properly designed analyses, and I was open to increasing my score. Unfortunately, the rebuttal has been quite disappointing, as the authors did not seem to make a serious effort to address these concerns, and their response feels perfunctory. I will decrease my score if the authors do not actively engage in the discussion.
> > >
> > > We are surprised and disheartened to hear that our efforts have been perceived as insufficient or perfunctory. We want to emphasize that we have put considerable time and effort into addressing all the concerns raised during the rebuttal phase. Specifically, we have trained two additional models to strengthen our ablation studies, implemented and evaluated an additional baseline (PanFusion), and evaluated all seven baselines on a completely new metric, which required significant time and computational resources. Furthermore, we have provided extensive out-of-distribution (OOD) results, updated the anonymous website multiple times, and added numerous qualitative results on critical aspects such as GroupNorm and overlap predictions.
> > >
> > > We have also made every effort to thoroughly address all reviewer questions, engaging in a detailed discussion to clarify our viewpoint on evaluating baselines. While it is possible to have differing perspectives on how baselines should be evaluated, we respectfully believe that such a difference in viewpoint should not be regarded as a lack of effort. Our responses were carefully crafted to provide a clear and thoughtful rationale.

---

> > > > ### Comment · Reviewer_oYCw · 2024-11-27
> > > >
> > > > ```
> > > > We are surprised and disheartened to hear that our efforts have been perceived as insufficient or perfunctory. We want to emphasize that we have put considerable time and effort into addressing all the concerns raised during the rebuttal phase. Specifically, we have trained two additional models to strengthen our ablation studies, implemented and evaluated an additional baseline (PanFusion), and evaluated all seven baselines on a completely new metric, which required significant time and computational resources. Furthermore, we have provided extensive out-of-distribution (OOD) results, updated the anonymous website multiple times, and added numerous qualitative results on critical aspects such as GroupNorm and overlap predictions.
> > > >
> > > > We have also made every effort to thoroughly address all reviewer questions, engaging in a detailed discussion to clarify our viewpoint on evaluating baselines. While it is possible to have differing perspectives on how baselines should be evaluated, we respectfully believe that such a difference in viewpoint should not be regarded as a lack of effort. Our responses were carefully crafted to provide a clear and thoughtful rationale.
> > > > ```
> > > >
> > > > I'm aware of your efforts on PanFusion and the new metric. However, it's true that you ignored the questions about implementation details and literature in Reviewer FHAm's initial review (maybe because Reviewer FHAm had a high initial score) and made claims like "having completely different training data doesn't lead to unfair experiments." IMHO, having a thorough ablation study is even more important than having one more baseline/metric, which can help others better understand your idea and inspire future research -- in the end, it's a research paper rather than a product

---

> > > > > ### Author Response · Authors · 2024-11-27
> > > > > **Response to Reviewer oYCw**
> > > > >
> > > > > Thank you for your feedback and for raising your score. We’re glad that you highlighted our ablation with limited data as a key point, as it strongly supports the effectiveness of cubemaps. This result demonstrates that even with minimal training data, our approach achieves strong performance, reinforcing our claims. We appreciate you pointing out the importance of these ablations early on and are glad they helped resolve your main concerns.
> > > > >
> > > > > We acknowledge that we missed addressing Reviewer FHAm's questions in the initial phase, but we have since made a significant effort to thoroughly answer all questions and address all concerns raised.
> > > > >
> > > > > Thank you again for your thoughtful engagement with our work.

---

> ### Author Response · Authors · 2024-11-26
> **Response to Reviewer oYCw and FHAm**
>
> Hi Reviewer oYCw and Reviewer FHAm,
>
> thank you for catching this error. We have updated the website to display the correct conditioning images, and we appreciate your attention to detail. We believe that the OOD examples strongly demonstrate the effectiveness of our method.
>
> Regarding your other points, we want to assure you that we are actively working on addressing them. We have been dedicating significant effort to this rebuttal, focusing on providing the additional experiments, evaluations, and results reviewers have asked for, e.g. evaluating another baseline and even an additional metric for alle baselines and our method. Currently, we are training the models you asked for with limited data and will share the results later today.
>
> We kindly ask for your patience, as training and evaluating these models take some time. Thank you again for your thorough feedback and understanding.
>
> Best regards
> The Authors

---

> ### Comment · Reviewer_FHAm · 2024-11-27
> **Reply to the discussions**
>
> I didn’t mean to start an argument—perhaps I miscommunicated. We are not trying to block your paper or ignore your efforts. I simply want to ensure that the value of your work is clearly visible through well-presented ablation studies and a detailed account of its history and your unique contributions. It was not very clear, but now it is getting better and better.

---

> > ### Author Response · Authors · 2024-11-27
> > **Response to Reviewer FHAm**
> >
> > Thank you for your clarification. We were just a bit surprised by the tone earlier but truly appreciate your feedback and engagement, which have definitely improved our submission. Thank you again for your valuable feedback! We are happy to hear that the submission is getting better and better :-)

---

### Official Review · Reviewer_MXUG · 2024-11-02

**Soundness:** 3
**Presentation:** 2
**Contribution:** 3
**Rating:** 8
**Confidence:** 4

**Summary:**

This work introduces CubeDiff, which is a simple and efficient method for generating 360 panorama from image and text conditions. CubeDiff builds on pretrained a multi-view diffusion model and fine-tunes it to output cubemap representations that is later stitched together into a single panorama. Compared to previous approaches that employ complex cross-view attention mechanisms, CubeDiff simply inflates existing attention layers and adds positional embeddings to adapt to cubemap representation. Thanks to this simple change to multi-view diffusion model, CubeDiff achieves high-quality panorama generation while retaining the generalizability of the base diffusion model. The qualitative and quantitative comparisons with baselines demonstrate the CubeDiff achieves better quality panoramas, and also enabling fine-grained text control which was challenging in previous methods.

**Strengths:**

* The proposed method is simple yet effective, according to the provided qualitative examples. Using cubemaps as a representation for panorama generation seems to be a valid choice, and building on a pretrained multi-view diffusion model is an adequate approach as it is proven to be effective at handling similar tasks requiring multi-view consistent generation.
* The method shows notable training efficiency, as only 48k panoramas were used for training. It is convincing that it adapts well to out-of-distribution conditions, as shown by the outdoor generations in Fig. 1.
* The fine-grainded text control of CubeDiff seems to be a clear advantage compared to previous approaches that use other types of panorama representatiions. From a user perspective, being able to provide certain controls for each of view would be more convenient with cubemaps, as it removes the need to consider the overlapping regions between neighboring views.

**Weaknesses:**

While the provided qualitative results of CubeDiff are convincing, several key aspects regarding the methodology and baseline comparisons are missing:

* In Sec. 4.1, it is remains unclear how the "inflated" attention layer in CubeDiff is different from previous appoaches, such as the Correspondence-Aware Attention from MVDiffusion [1]. Specifically, it's unclear why this method reduces the risk of overfitting and results in greater generalization compared to approaches like MVDiffusion. This is particularly confusing since MVDream [2], from which CubeDiff adapts its layers, also incorporates an additional "3D Self-Attention" layer on top of a 2D diffusion model. Further clarification would be helpful to understand why one approach may lead to overfitting (as in MVDiffusion), while the other leads to generalization (as in CubeDiff).

* Missing baseline: PanFusion [3] is an imporant baseline to consider for 360 panorama generation from text prompts. As it was published in CVPR 2024, it serves as a valid baseline for this work. To claim state-of-the-art performance on panorama generation, comparisons with PanFusion (e.g. in terms of image quality, generalization) would be required. Alternatively, it would be helpful to provide explanations on how CubeDiff differs from PanFusion and why a direct comparison may not be necessary.

* I am curious about the details of the quantative evaluation procedure for the FID and KID metrics. Were the generated 360 panoramas directly used for measuring FID, or were projected perspective views used instead? Additionally, there exists a variant of FID, called FAED [4], specifically designed to assess the quality and realism of panoramas. Evaluating with such metrics could provide a more accurate reflection of the actual quality of the panorama generations.

* I could not find ablation studies for the key components of the method, specifically regarding the effectiveness of synchronized group normalization and overlapped predictions. These are critical design choices for CubeDiff, yet it is difficult to assess their validity without quantative or qualitative comparisons showing the impact of these components.

Overall, I agree that the cubemap representation could be an effective choice for panorama generation, and CubeDiff introduces a new approach of incorporating the cubemap representation. However, without the concerns about the key contribution addressed - particularly a clear explanation on why this approach should lead to enhanced generalization to out-of-distribution cases - it is difficult to assign a higher rating.

[1] MVDiffusion: Enabling holistic multi-view image generation with correspondence-aware diffusion, Tang et al., NeurIPS 2023

[2] MVDream: Multi-view Diffusion for 3D Generation, Shi et al., ICLR 2024

[3] Taming Stable Diffusion for Text to 360 Panorama Image Generation, Zhang et al., CVPR 2024

[4] Bips: Bi-modal indoor panorama synthesis via residual depth-aided adversarial learning, Oh et al., ECCV 2022

**Questions:**

The main questions on the method are in the Weaknesses section.

* While fine-grained text control appears useful, I'm curious whether the conditioning can truly be applied on a "per-face" basis. Interactions between the pixels in different views may influence the content generated in each other. Could there be failure cases where changing the text condition for one view can lead to unintended changes in another view?

* As described in Sec. 4.5, CubeDiff is trained with both dropped text and image conditions. In this case, is CubeDiff also capable of performing text-conditioned generation without an image input?

**Details Of Ethics Concerns:**

Ethics review doesn't seem to be required for this work.

---

> ### Author Response · Authors · 2024-11-22
> **Response to Reviewer MXUG**
>
> We thank Reviewer MXUG for the detailed review. In the following, we try to assess all questions and concerns.
>
> **Regarding the first point:**
>
> This is indeed an important aspect of our paper, thank you for giving us the opportunity to clarify:
>
> **Differences to MVDiffusion**: MVDiffusion introduces a correspondence-aware attention mechanism, which, while innovative, has notable limitations. Specifically, it operates only within a very limited receptive field (a local neighborhood of 9), and ia initialised with all zeros. This means that attention weights must be trained from scratch, consequently they are prone to overfitting and do not generalize all that well. In contrast, the attention mechanism of CubeDiff spans all six frames at their full spatial extent: each latent can attend to every other latent in the feature map, leading to a a much more comprehensive contextual understanding. What is more, for CubeDiff we only adjust the token length while preserving the channel size, which means that it can exploit pretrained weights learned from internet-scale data. We point the reviewer to the ablation studies in the MVDream paper, which show that inflating the 2D self-attention layer to 3D significantly reduces overfitting and improves generalization, overcoming a main design limitation of MVDiffusion.
>
> **Comparison with MVDream**: MVDream explores three different architectural strategies, including the addition of a separate 3D self-attention layer on top of the existing 2D self-attention layer. However, this additional layer must again be trained from scratch rather than leverage pretrained weights, leading to overfitting. The ablation studies in the MVDream paper demonstrate that, to inflate the 2D self-attention layer to 3D context, it is better to extend the token sequence length, such that pretrained weights can be used. For CubeDiff we adopt this approach. Given that our choice is informed by the earlier findings of the MVDream authors we chose not to repeat this specific ablation study.
>
> **Regarding the second point:**
>
> Thank you for bringing PanFusion to our attention. We agree that it is an important baseline for panorama generation from text prompts, and have therefore added it to our study. As shown in the revised Tables 1, CubeDiff surpasses PanFusion's performance in 9 out of 10 metrics, including the proposed FAED metric. We also include qualitative comparisons in the revised paper and on our anonymous website to visually demonstrate the superior image quality and generalization of CubeDiff.
>
> We note that CubeDiff differs from PanFusion in several key aspects. PanFusion employs a two-branch pipeline with both a panorama model and a perspective model. To allow cross-branch communication, the authors develop a custom attention module that links equirectangular and perspective projections, which once again must be trained from scratch. It is an important design choice of CubeDiff to set up the architecture such that the model can build on broadly applicable, "foundational" attention weights learned through large-scale pretraining.
>
> **Regarding the third point:**
>
> Thank you for pointing out the FAED metric, and for the question regarding the FID and KID metrics, this is indeed an important detail.
>
> **FID, KID, and CLIP-FID Evaluation**: We followed a similar protocol as existing works and measure them on 10 random perspective views extracted from each generated 360° panorama (excluding the input image). In this way, the metrics are computed on the expected image format and allows for a fair comparison across models.
>
> **FAED Evaluation**: We evaluated FAED using PanFusion’s autoencoder pretrained on Matterport3D. As shown in Table 1, CubeDiff consistently outperforms other models in FAED, highlighting its ability to generate realistic, high-fidelity panoramas.
>
> **Regarding the fourth point:**
>
> We agree that we should have better supported the decision to employ synchronized group normalization and overlapping predictions. We have now included qualitative results in the manuscript to demonstrate the impact of these components.
>
> Quantitative comparisons for these components are challenging: their main benefit is to prevent subtle color shifts and discontinuities at the border of cube faces, but existing metrics (including FAED) are not sensitive to these artifacts; whereas they stand out for human perception and markedly degrade visual quality. Hence, visual inspection offers a more meaningful insight into the effects of these components. More results can be found in the appendix.
>
> **Overall**:
>
> We are glad the reviewer recognizes the novelty and potential of our cubemap-based approach. CubeDiff’s inflated attention mechanism is key to its performance, enabling spatial coherence and leveraging generalizable, pretrained attention weights. We revised the manuscript to emphasize these points and better explain CubeDiff’s strengths. We hope our revisions address the reviewer’s concerns.

---

> > ### Comment · Reviewer_MXUG · 2024-11-23
> >
> > I appreciate the authors for providing detailed explanations on the design choice behind "inflated attention". This makes it more clear why this approach leads to better generalization compared to MVDiffusion.
> >
> > Also, provided the comparisons with PanFusion and experiments on FAED metric, and the ablations, it becomes clear that CubeDiff outperforms prior works.
> >
> > Overall, I agree that the cubemap representation is a promising choice for high-quality panorama generation, and this paper suggests a straightforward yet effective method to incorporate cubemaps based on 2D diffusion models. Given that, I'd like to raise my rating to 8.
> >
> > Lastly, while MVDream demonstrated that extending the token sequence is more effective, it would be nice if the authors include at least a toy experiment in the final version to show that the same holds for the cubemap representation as well. It would strengthen the argument in the paper.

---

> > > ### Author Response · Authors · 2024-11-23
> > > **Response to Reviewer MXUG**
> > >
> > > Thank you for your feedback. We are glad to hear that our explanations and comparisons helped clarify the strengths of our approach and its contributions to panorama generation. We will make sure to include the additional ablation in the final version of the paper. Thank you for taking the time to review our work and for your constructive comments. We are grateful for raising your rating.

---

### Meta-Review · Area_Chair_eMDK · 2024-12-22

**Metareview:**

The paper received very favorable scores from the reviewers. The reviewers praise simplicity, impressive results (even OOD ones), sufficient discussion and experiments. The reviewers also had a number of questions, which the authors did a good job addressing during the discussion period. AC checked the paper, discussion, visual results and agrees with the reviewers. Congrats!

**Additional Comments On Reviewer Discussion:**

There was a very healthy discussion, during which new clarifying information was added. Due to the request mPic, the authors added OOD results to the supplement website. The results look convincing. Reviewers oYCw and FHAm engaged in a discussion about the missing ablations, which the authors tried addressing.

---

### Decision · Program_Chairs · 2025-01-22

Accept (Spotlight)